# How Many Tokens Do 3D Point Cloud Transformer Architectures Really Need?

**Tuan Anh Tran**[1]**, Duy M. H. Nguyen**[1,2,3]**, Hoai-Chau Tran**[4,5]**, Michael Barz**[1]**,
Khoa D. Doan**[4,5]**, Roger Wattenhofer**[6]**, Ngo Anh Vien**[5,7]**, Mathias Niepert**[2,3]**,
Daniel Sonntag**[1,8]**, Paul Swoboda**[9]

[1]German Research Centre for Artificial Intelligence (DFKI)
[2]Max Planck Research School for Intelligent Systems (IMPRS-IS)
[3]University of Stuttgart, [4] VinUni-Illinois Smart Health Center, VinUniversity
[5]College of Engineering & Computer Science, VinUniversity, [6]ETH Zurich,
[7]VinRobotics, [8]University of Oldenburg, [9]Heinrich Heine University Düsseldorf
{tuan.tran, ho_minh_duy.nguyen}@dfki.de, paul.swoboda@hhu.de

## Abstract

Recent advances in 3D point cloud transformers have led to state-of-the-art results in tasks such as semantic segmentation and reconstruction. However, these models typically rely on dense token representations, incurring high computational and memory costs during training and inference. In this work, we present the finding that tokens are remarkably redundant, leading to substantial inefficiency. We introduce **GitMerge3D**, a **g**lobally **i**nformed graph **t**oken **merging** method that can reduce the token count by up to 90–95% while maintaining competitive performance. This finding challenges the prevailing assumption that more tokens inherently yield better performance and highlights that many current models are over-tokenized and under-optimized for scalability. We validate our method across multiple 3D vision tasks and show consistent improvements in computational efficiency. This work is the first to assess redundancy in large-scale 3D transformer models, providing insights into the development of more efficient 3D foundation architectures. Our code and checkpoints are publicly available at https://gitmerge3d.github.io.

## 1 Introduction

The rise of transformer-based architectures has significantly advanced the field of 3D point cloud understanding [35, 101, 28], particularly in tasks such as semantic segmentation [47, 87, 97, 46], object detection [25, 36, 57, 92], and reconstruction [44, 11, 80, 14]. Building upon the success of attention mechanisms in natural language processing [82, 19, 1] and 2D vision [20, 9, 54, 42], early works like Point Transformer (PTv-1) [102] introduced self-attention tailored to the irregular and unordered nature of point clouds. This line of research evolved through subsequent versions, PTv-2 [93] and the more scalable PTv3 [92], which incorporated enhancements such as local-global feature fusion, serialized points ordering and optimized positional encoding. Among these, PTv3 has emerged as a particularly powerful backbone, capable of capturing complex spatial dependencies and scaling effectively to large-scale scenes. Leveraging large-scale attention mechanisms based on a 1D serialized ordering of 3D points and a hierarchical architecture, it enables rich geometric reasoning and excels in dense 3D semantic segmentation tasks [27, 91], where fine-grained boundary understanding is essential. PTv3 also serves as a strong encoder in advanced 3D reconstruction pipelines, such as SplatFormer [14] or LSM [27], and excels in 3D object detection [31, 84, 61] by effectively capturing both local and global features. These strengths make PTv3 a key enabler of high-performance and generalizable solutions in modern 3D vision.

39th Conference on Neural Information Processing Systems (NeurIPS 2025).

While PTv3 has emerged as a cornerstone for high-performance and generalizable 3D vision models, our analysis reveals a surprising inefficiency at the core of its attention mechanism. Despite architectural advancements aimed at improving scalability, such as replacing computationally expensive K-nearest neighbor operations (28% of inference time) with lightweight 1D serialized neighbor mappings and removing image-relative positional encodings (26% of inference steps) to efficiently expand the receptive field from 16 to 1024 points, PTv3 still dramatically overutilizes tokens during self-attention. *Strikingly, we show that preserving only 5–10% of the most spatially informative tokens is sufficient to maintain nearly identical performance across diverse 3D tasks*. This challenges the prevailing belief that dense tokenization is essential for transformer performance in 3D domains [34, 102, 56, 45]. To our knowledge, this is the first work to reveal and systematically analyze the redundancy in token usage, pointing to significant opportunities for improving the efficiency of point cloud transformers without compromising accuracy (Figure 1).

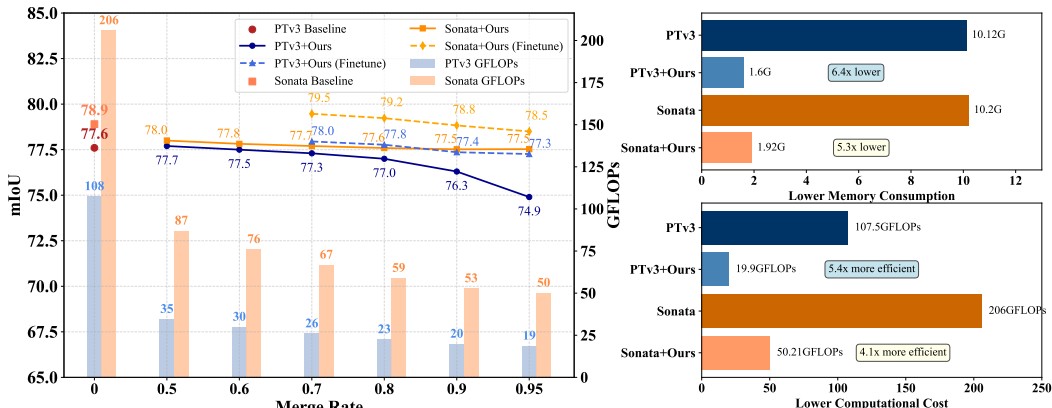

Figure 1: We compare the original PTv3 and Sonata with our proposed token merging method (PTv3 + Ours and Sonata + Ours) in terms of FLOPs and memory consumption. Despite merging up to 90% of tokens, our method applied to PTv3 achieves a **5.3 x reduction in FLOPs** (from 107.5 GFLOPs to 19.9 GFLOPs) and a **6.4 x reduction in memory usage** (from 10.12 GB to 1.6 GB), with minimal performance degradation. Notably, the model maintains comparable accuracy when fine-tuned by updating only the MLPs before and after the attention layer for just 10% of the original training epochs, while requiring significantly less computation per epoch during fine-tuning.
.

In particular, we conducted a comprehensive set of experiments by integrating several prominent token reduction techniques, originally developed for vision transformers, into the PTv3 framework. These included Token Merging (ToMe) [5–7], Token Pruning [99, 103, 41], ALGM [66], and PiToMe [81]. We inserted these methods before each attention layer to merge or prune varying proportions of tokens (from 10% to 50%) *during inference*, followed by an unmerging step to restore the token structure for compatibility with subsequent MLP layers. Remarkably, across the standard PTv3 and its advanced variants, such as PTv3 Sonata [91] and Splatformer [14], we observed negligible accuracy degradation even under substantial token reduction. This surprising resilience held consistently across a range of benchmarks, including 3D semantic segmentation (ScanNet [16], S3DIS [2], nuScenes [8]), novel view object reconstruction (ShapeNet [10], Objaverse [18]), and object detection (Waymo [77]), while significantly reducing both FLOPs and memory consumption.

Encouraged by this observation, we hypothesized that a domain-specific merging mechanism, one that explicitly accounts for spatial locality and attention relevance in 3D point clouds, could unlock even greater efficiency. To this end, we developed a novel 3D-aware token merging strategy capable of merging up to 99% of tokens without retraining. The results are striking: even with 95% of tokens removed, our method maintains competitive performance while pushing computational and memory efficiency to new bounds (see Figure 1). Furthermore, with just 10% of the original training schedule allocated for fine-tuning, the model not only fully recovers its baseline performance but even surpasses it in some cases, e.g., in ScanNet or S3DIS datasets, highlighting the practical potential of our method for scalable and efficient real-world deployment.

In summary, we make the following contributions:

- Through a systematic study, we uncover a surprising degree of token redundancy in state-of-the-art point cloud transformers, showing that up to 90-95% of tokens can be removed without significant performance drop. This challenges the common assumption that dense tokenization is essential for 3D transformer effectiveness.

- We propose a 3D-specific token merging strategy that integrates local geometric structure and attention saliency to estimate voxel importance, enabling aggressive token reduction with minimal accuracy degradation.

- We validate the proposed token merging across 3D semantic segmentation, reconstruction, and detection tasks, achieving substantial efficiency gains and, in some cases, surpassing the baseline performance with minimal fine-tuning. We expect these findings will pave the way for future research toward lightweight and scalable transformer architectures for 3D point cloud processing.

## 2 Related Work

**3D Point Cloud Architectures.** 3D point cloud understanding has evolved through multiple architectural paradigms. Early approaches include projection-based methods [13, 48, 49, 76], which project point clouds onto 2D image planes for processing with standard CNNs, and voxel-based methods [63, 75, 15, 32, 88], which discretize space into regular grids to apply 3D convolutions. While effective, these techniques often suffer from resolution loss, high memory consumption, or limited geometric expressiveness. To address these challenges, point-based methods such as PointNet [67], PointNet++ [68], and the more recent PointMLP [60] directly operate on raw point sets, preserving fine-grained spatial structure. However, their reliance on local operations can still limit global context modeling. This has led to a growing shift toward transformer-based architectures that better capture long-range dependencies in 3D data. The Point Transformer (PTv) family, spanning PTv-1 [102], PTv-2 [93] to the more scalable PTv3 [92], adapts attention mechanisms to unordered point sets and has become a state-of-the-art backbone for 3D tasks such as semantic segmentation [47, 87, 97, 46], object detection [25, 36, 57, 92], and reconstruction [44, 11, 80, 14]. Building on PTv3's success, recent variants like PTv3 Sonata [91], pretrained on 140K point clouds for improved generalization, and Splatformer [14], tailored for robust 3D novel view synthesis, further demonstrate the versatility and dominance of transformer-based models in modern 3D vision pipelines.

In contrast to prior works, our study takes a step back to ask a fundamental question: *"Is PTv3 already efficient in how it uses tokens?"* Our findings reveal that the model can be significantly compressed, preserving only a fraction of tokens while maintaining comparable accuracy, opening a new direction for building lightweight and memory-efficient 3D point cloud transformers.

**Token Redundancy and Sparsity in Transformers.** To improve transformer efficiency, prior work has explored a range of strategies, including approximating attention via hashing [17, 43], low-rank factorization [52, 26], or sparsity [71, 74], as well as head pruning [64, 29] and domain-specific modules [54, 55, 89]. While effective, many of these methods require retraining or extensive fine-tuning from scratch, limiting their practicality. In contrast, token reduction techniques such as token pruning [99, 103, 85, 41] and token merging [5–7, 81] aim to accelerate inference by reducing the number of tokens processed, often with minimal accuracy degradation. Notably, methods like ToMe [5] and its variants [6, 66] leverage bipartite soft matching to efficiently merge similar tokens, though they may suffer from heuristic decisions or sensitivity to token distributions. Other approaches employ clustering [4, 62] or graph-based methods [85, 96, 81] to merge tokens more systematically, but these often introduce computational overhead that counters their intended efficiency. In this work, we systematically adapt several general-purpose token reduction methods to PTv3 and its advanced variants, uncovering that these models preserve performance even after substantial token reduction during inference. Motivated by this insight, we introduce a 3D-specific token merging strategy that incorporates spatial structure and densities among regions in 3D scenes, which enables up to 99% token reduction while achieving substantial efficiency gains with minimal or no performance loss, pointing to a promising direction for scalable 3D transformer design.

**3D Point Cloud Compression and Efficiency.** There is a line of work that focuses on designing efficient architectures for 3D point clouds, such as MinkUNet [15], Sparse Point Transformer [79, 78, 86], PTv3 [92], and others [47, 30, 39], which significantly reduce computation through

architectural innovations. However, these models typically *require training from scratch, which limits their adaptability and prevents seamless integration with pre-trained models.* In parallel, various off-the-shelf techniques have been proposed to reduce the size of 3D point clouds before feeding them into neural networks, including Random Sampling [37, 58, 100], Farthest Point Sampling (FPS) [24, 68, 51, 50], and VoxelGrid Downsampling [70, 59, 83]. While these methods are simple and effective at reducing input size, they are typically rule-based and insensitive to the underlying task or feature importance, leading to suboptimal performance in downstream applications. In contrast, our token merging strategy operates at the feature level and is computationally flexible, which allows dynamic token compression during inference *without retraining* and integrates seamlessly with existing architectures like PTv3. Furthermore, we consistently outperform traditional downsampling methods in both efficiency and performance across segmentation, reconstruction, and detection tasks.

## 3 Analyzing Token Redundancy in 3D Transformers

### 3.1 Point Transformer v3 architecture

PTv3 introduces a simplified and efficient framework for 3D point cloud processing by replacing KNN-based grouping with a 1D serialization strategy, where points are ordered via space-filling curves to preserve spatial locality. The model follows a U-Net-style encoder-decoder architecture with skip connections, enabling hierarchical feature learning (Figure 3).

At each resolution, the serialized sequence is partitioned into disjoint local groups, and *self-attention* is applied independently to each group to capture local context. PTv3 evenly divides the input point set $\mathcal{X} = \{x_1, x_2, \ldots, x_N\}$ into $K$ disjoint subsets (partitions) $\{\mathcal{P}_1, \mathcal{P}_2, \ldots, \mathcal{P}_K\}$ such that $\bigcup_{k=1}^{K} \mathcal{P}_k = \mathcal{X}, \mathcal{P}_i \cap \mathcal{P}_j = \varnothing$ for $i \neq j$, and $|\mathcal{P}_k| = 1024$. Self-attention is then applied independently within each partition to capture local geometric structure.

The attention for each point $x_i$ is computed only over the points in its own partition $\mathcal{P}(i)$, formulated as:

$$\text{Attn}(x_i) = \sum_{x_j \in \mathcal{P}(i)} \text{softmax}_j \left( \frac{\mathbf{q}_i^\top \mathbf{k}_j}{\sqrt{d}} \right) \mathbf{v}_j,$$

where $\mathbf{q}_i = \mathbf{q}(x_i), \mathbf{k}_j = \mathbf{k}(x_j)$, and $\mathbf{v}_j = \mathbf{v}(x_j)$ denote the query, key, and value projections of the respective points. The summation is restricted to $x_j \in \mathcal{P}(i)$, ensuring attention is confined to the local context defined by the partition. While this attention mechanism offers strong representational capacity, it becomes prohibitively slow and computationally expensive with $\mathcal{O}(N^2)$ complexity, especially when processing point clouds containing millions of points.

### 3.2 Token Merging Formulation

To address this limitation, *token merging [5, 81, 12, 66]* is introduced to reduce the number of tokens participating in the attention computation. Each original token is mapped to a merged representation via a function $f : x_i \mapsto \widetilde{x}_i$, inducing a transformation of the attention partition $\mathcal{P}(i) \rightarrow \widetilde{\mathcal{P}}(i)$, where $|\widetilde{\mathcal{P}}(i)| < |\mathcal{P}(i)|$. Attention is then computed over the merged tokens as:

$$\text{Attn}(f(x_i)) = \sum_{\widetilde{x}_j \in \widetilde{\mathcal{P}}(i)} \text{softmax}_j \left( \frac{f(\mathbf{q}_i)^\top f(\mathbf{k}_j)}{\sqrt{d}} \right) f(\mathbf{v}_j),$$

The function $f$ merges token features according to a learned token-level mapping. Unlike existing token merging methods [5, 81, 12, 66], which are designed for classification and operate on the merged tokens throughout subsequent layers, **our approach targets dense 3D point cloud processing**. This requires restoring the token features to their original resolution. To achieve this, an unmerging function $f^{-1}$, approximating original attention layer is required: $\text{Attn}(x_i) \approx f^{-1} \left( \text{Attn}(f(x_i)) \right).$

**Token Merging.** Token Merging (ToMe) [5] defines the merge function $f(\{x_i\}, r)$ by partitioning the set of tokens $\{x_i\}$ into source (src) and destination (dst) sets, and assigning the $r$ most similar tokens from src to tokens in dst. Each merged token is computed by averaging the features of its assigned source tokens along with the corresponding destination token.

Besides ToMe, we also evaluate several existing token merging methods, including PiToMe and ALGM, on two 3D segmentation benchmarks: ScanNet and ScannNet200, using PTv3 and PTv3

Sonata. For each method, we apply different token merging ratios and report both GFLOPs and mIoU.

As shown in Fig. 1 and Fig. 2, we observe that model performance remains stable, i.e., mIoU drops only slightly - even when up to 50% of tokens are merged. Meanwhile, GLOPs are reduced significantly, demonstrating the redundancy in token representations. This suggests that aggressive token reduction is feasible for 3D data.

However, because existing methods are designed primarily for image classification and are not optimized for the characteristics of 3D point clouds. Their merging strategies are generic and do not exploit spatial locality or the density variation in 3D scenes. Moreover, most of them do not support feature recovery, which is crucial for dense segmentation. This motivates us to propose a new token merging method specifically designed for 3D point clouds. Our approach allows higher merging ratios while preserving fine-grained information necessary for accurate segmentation. It integrates a learned merging and unmerging mechanism to reduce attention cost while maintaining per-point predictions.

# 4 Aggressive Merging Methods for Point Clouds Processing

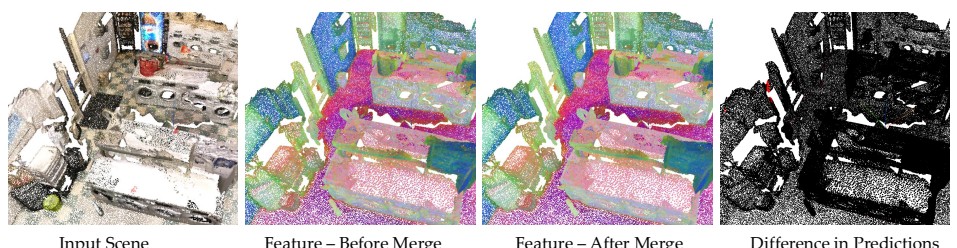

| Input Scene | Feature – Before Merge | Feature – After Merge | Difference in Predictions |

Figure 2: **Observation**: After merging 90% of the tokens in each attention layer, the change in PCA visualization of feature representation (3rd image) is minimal compared to the original feature (2nd image). Most of the predictions remain unchanged after merging, with red indicating the areas where predictions differ. This leads us to conclude that there is high redundancy in the point cloud processing model.

**Empirical Observations.** As shown in Figure 3, when applying spatial-preserving token merging at extreme rates (e.g., 90% and 95%), the majority of points retain their original predictions. Additionally, PCA visualizations of token features reveal that, in high-resolution layers of PTv3, features are well-separated by object, indicating strong spatial and semantic consistency.

**Adaptive Merging via Global-Informed Energy Score.** We observe that aggressive merging (e.g., >90%) can still retain performance if the merging ratio is adapted to the information content of each partition. Inspired by the energy score in existing token merging methods [81, 5, 66, 12], we propose a global-informed energy score to guide adaptive merging decisions.

We define a bipartite graph $G = (\mathcal{V}, \mathcal{E})$, where the vertex set is $\mathcal{V} = \{x_i\} \cup \{\bar{P}_j\}$. Here, $\bar{P}_j$ denotes the centroid of partition $\mathcal{P}_j$, computed as: $\bar{P}_j = \frac{1}{|\mathcal{P}_j|} \sum_{x_k \in \mathcal{P}_j} x_k$. The edge set is defined as $\mathcal{E} = \{(x_i, \bar{P}_j)\}$, forming a directed bipartite graph from each token $x_i$ to all partition centroids $\bar{P}_j$. For each token $x_i$, we define its outgoing neighbors as $\mathcal{N}(x_i) = \{\bar{P}_j \mid (x_i, \bar{P}_j) \in E\}$.

The energy score $E(x_i)$ is then computed as the mean cosine similarity between $x_i$ and all connected centroids:

$$E(x_i) = -\frac{1}{|\mathcal{N}(x_i)|} \sum_{\bar{P}_j \in \mathcal{N}(x_i)} \cos(x_i, \bar{P}_j). \tag{1}$$

This score reflects how globally aligned a token is with all partition centroids. Tokens with lower energy (i.e., more aligned with global structure) are considered less informative and can be merged more aggressively, whereas high-energy tokens (i.e., less aligned with global structure) are preserved to retain critical information.

**Adaptive Merging by Energy.** Using the above formulation, we define the importance score of a partition $\mathcal{P}$ as the mean energy of its tokens:

$$E(\mathcal{P}) = \frac{1}{|\mathcal{P}|} \sum_{x \in \mathcal{P}} E(x).$$

If $E(\mathcal{P}) > \tau$, we apply moderate merging $f(\mathcal{P}, r)$; otherwise, we apply aggressive merging $f(\mathcal{P}, r^+)$, where $r^+ \gg r$. This branching mechanism enables us to aggressively reduce redundancy and significantly improve computational efficiency, while still supporting batch training and preserving performance on off-the-shelf evaluation. Here $\tau$ is a common threshold we effectively used for all datasets and tasks. We present in Figure 3 our proposed algorithm with further insights in Sec. D, Appendix.

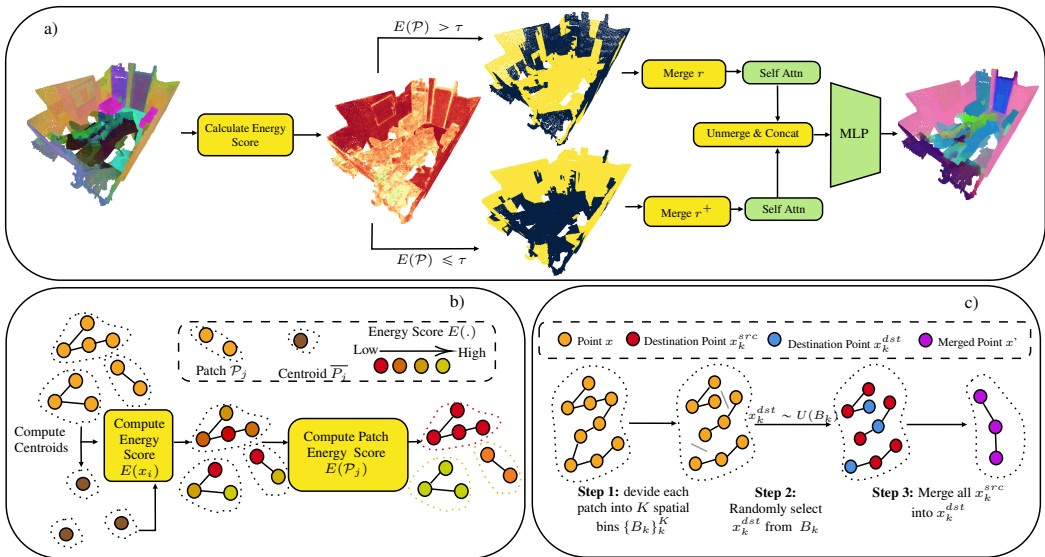

Figure 3: **b)** For each Point Transformer layer, we compute token energy scores and propagate them to patches using a globally informed graph over the local self-attention. **a)** These patch-level scores guide adaptive merging, retaining more information for high-energy patches. **c)** Each patch is divided into evenly sized bins, and destination tokens are randomly selected within these bins to enable spatially aware merging.

## 5 Experimental Results

### 5.1 Downstream Tasks and Baseline Setup

We evaluate our method on three 3D tasks: **3D Semantic Segmentation**, **3D Reconstruction**, and **Object Detection**. For semantic segmentation, we test our approach on Sonata [91] and PTv3 [92] across four datasets: ScanNet200 [72], ScanNet [16], S3DIS [2], and NuScenes [8]. For 3D reconstruction, we evaluate our method using SplatFormer [14] on three datasets: ShapeNet [10], ObjectVerse [18], and GSO [22]. We present results on the evaluation sets of indoor semantic segmentation datasets, while test set results and an additional **object detection** task are assessed using the state-of-the-art language-guided object detector SpatialLM [61].

In addition to recent token merging methods [81, 5, 66], we incorporate point cloud downsampling techniques to reduce input complexity. **Random Token Drop** [37] randomly discards a subset of points, offering fast but coarse reduction. **Farthest Point Sampling** (FPS) [21, 95] selects points that are maximally distant from each other to preserve geometric coverage. **VoxelGrid Downsampling** [70, 59] partitions space into voxels and retains one representative point per voxel, ensuring spatial regularity. Final predictions are upsampled in the last stage.

### 5.2 3D Point Cloud Semantic Segmentation

**Indoor Segmentation:** We evaluate our method using GFLOPs and mIoU on Sonata and PTv3 (Fig.4). Even without fine-tuning, our approach shows minimal segmentation drop. Finetuning

with only 10% of the original training epochs , our merging strategy significantly outperforms others in efficiency. At 80% merging for high-energy and 97% for low-energy branches (K=32), performance remains unaffected. Table 1 also shows that our method outperforms traditional point cloud downsampling techniques by preserving more latent information, leading to better results.

As shown qualitatively in Tab. 2, even with up to 95% token merging, feature representations remain largely unchanged, indicating high redundancy and supporting our aggressive merging strategy. We attribute this to the nature of point cloud data, which is both sparse in 3D space and fine-grained. Our finding suggests a more efficient way to process such data.

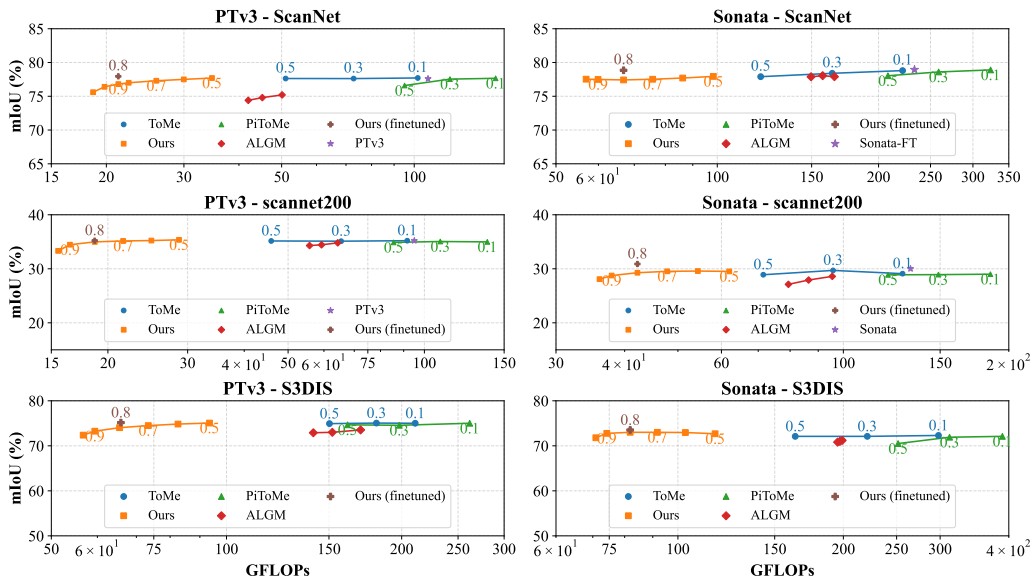

Figure 4: Off-the-shelf performance comparison between our merging against existing methods on PTv3 Sonata and PTv3 across three datasets ScanNet, ScanNet-200 and S3DIS. The numbers above each data point indicate the merging rate.

Table 1: We compare our method, using a merge rate of 0.8, in two settings- fine-tuned (blue rows) and off-the-shelf (gray rows) - against other segmentation and point cloud downsampling methods applied to PTv3.

| Methods | ScanNet Val | ScanNet200 Val | S3DIS Area5 |
|---|---|---|---|
| MinkUNet [15] | 72.2 | 25.0 | 65.4 |
| ST [47] | 74.3 | - | 72.0 |
| PointNeXt [69] | 71.5 | - | 70.5 |
| OctFormer [87] | 75.7 | 32.6 | - |
| Swin3D [97] | 76.4 | - | 72.5 |
| PTv1 [102] | 70.6 | 27.8 | 70.4 |
| PTv2 [93] | 75.4 | 30.2 | 71.6 |
| PTv3 [92] | 77.6 | 35.2 | 74.7 |
| *- Random Drop* | 70.1 | 31.1 | 73.4 |
| *- FPS* | 71.2 | 32.4 | 70.9 |
| *- VoxelGrid Down.* | 72.1 | 32.2 | 69.1 |
| - Ours | 77.0 | 34.4 | 72.3 |
| - Ours | 77.4 | 35.2 | 74.3 |
| PTv3-Sonata [91] | 79.0 | 30.4 | 72.2 |
| *- Random Drop* | 72.2 | 25.2 | 68.5 |
| *- FPS* | 73.9 | 26.1 | 69.0 |
| *- VoxelGrid Down.* | 73.9 | 25.5 | 68.8 |
| - Ours | 77.5 | 28.8 | 72.8 |
| - Ours | 78.9 | 30.9 | 73.5 |

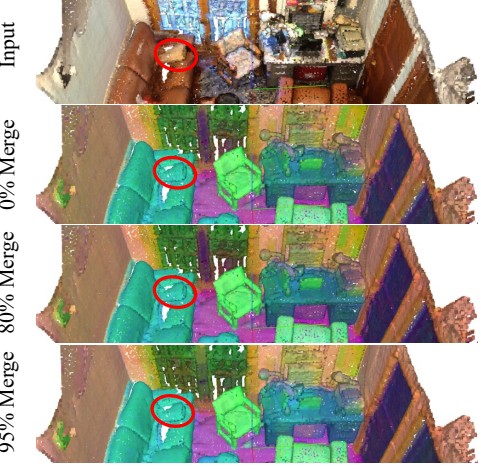

Table 2: PCA feature visualization of PTv3 on the 3D indoor segmentation task across different merging rates. Even at a 95% merging rate, the latent representations of the point cloud remain largely unchanged.

**Outdoor Segmentation:** We additionally evaluate our method on the outdoor dataset NuScenes [8]. Table 3 summarizes the performance and efficiency of different methods on the NuScenes validation

set in terms of mIoU, mAcc, allAcc, peak memory usage, GFLOPS, and latency. We use the default merging rate of 80% and an aggressive merge configuration with $K = 32$. PTv3 achieves the highest overall accuracy with an mIoU of 80.3, mAcc of 87.2, and allAcc of 94.6. While PTv3 + Ours shows slightly lower mIoU (78.0) and mAcc (85.5), it maintains a comparable allAcc (94.0), demonstrating minimal performance loss despite substantial computational savings. Notably, our method reduces peak memory usage by over 85% (from 6.20 GB to 0.92 GB), lowers GFLOPS by nearly 70%, and cuts latency by about 30%, highlighting significant improvements in speed and resource efficiency.

| Methods | mIoU | mAcc | allAcc | peakMem (GB) | GFLOPS | Latency (ms) |
|---|---|---|---|---|---|---|
| PTv2 [93] | 80.2 | - | - | - | - | - |
| PTv3 [92] | 80.3 | 87.2 | 94.6 | 6.20 | 101.68 | 152 |
| PTv3 + Ours | 78.0 | 85.5 | 94.0 | 0.92 | 32.45 | 106 |

Table 3: Comparison of semantic segmentation performance and efficiency on the NuScenes validation set [8].

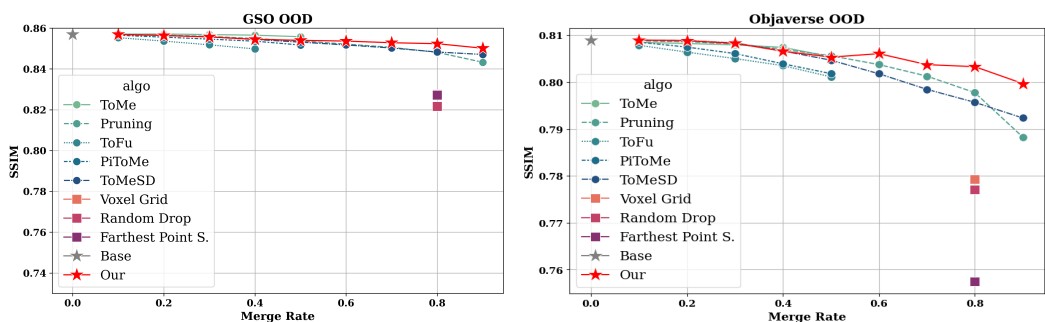

Figure 5: **3D Object reconstruction**: Off-the-shell performance of MAYC on Objaverse [18] and Google Scanned Object (GSO) [23].

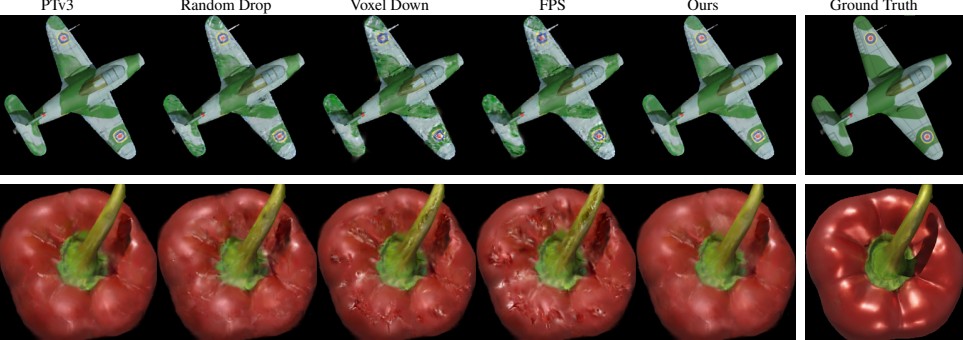

Figure 6: We visualize the output of various token compression techniques after removing 80% of the tokens, comparing their visual quality degradation (or preservation) on the 3D object reconstruction task.

## 5.3 3D Object Reconstruction

We also conduct experiments to evaluate the performance of our method on the novel view synthesis task under out-of-distribution (OOD) test camera angles. For this task, we adopt SplatFormer [14] as the backbone, which also incorporates PTv3 as its core to refines flawed 3D Gaussian splats to mitigate artifacts in OOD views.

As shown in Table 6, Figure 5 and 6, our method archives high performance, with only about a 0.1% drop across all metrics, even after reducing up to 90% of the tokens processed by the model, while still outperforming other state-of-the-art methods such as MipNeRF360 [3], 3DGS [40], 2DGS

[38], Nerfbusters [90], and LaRA [11]. In contrast, alternative token compression techniques such as Random Drop, Voxel Downsampling, and Furthest Point Sampling significantly degrade model performance after reducing 80% number of tokens.

## 5.4 Language Guided Object Detection

| Methods | F1 L25 | F1 L50 | F1 O25 | F1 O50 | Time (s) | Mem (GB) |
|---------|--------|--------|--------|--------|----------|----------|
| SpatialLM | 0.4906 | 0.3886 | 0.3356 | 0.1894 | 6.009 | 12.36 |
| + Ours (r=0.5) | 0.4982 | 0.3806 | 0.3490 | 0.1946 | 5.269 | 3.75 |
| + Ours (r=0.8) | 0.4809 | 0.3873 | 0.3485 | 0.2006 | 4.795 | 2.53 |

Table 4: Evaluation of our method with off-the-shelf setting on the SpatialLM dataset for object detection at two merging rates (0.5 and 0.8) demonstrates improvements in inference time and memory usage without any degradation in prediction quality.

SpatialLM is a language-guided 3D object detection model that uses Sonata as its backbone. It processes 3D point clouds to detect spatial layouts and objects based on natural language instructions.

We extensively evaluate our method on the language-guided object detection task from SpatialLM [61], which is based on Sonata. The metrics reported in Table 4 capture various aspects of detection performance and computational efficiency. Specifically, the F1 scores for layouts and objects at 25% and 50% Intersection-over-Union (IoU) thresholds (F1 L25, F1 L50, F1 O25, F1 O50; where L refers to scene layouts and O to objects) assess the accuracy of detecting spatial layouts and objects with different levels of strictness. Inference time (in seconds) and peak memory consumption (in GB) reflect the computational cost during model inference. Figure 7 shows that even after merging 80% of

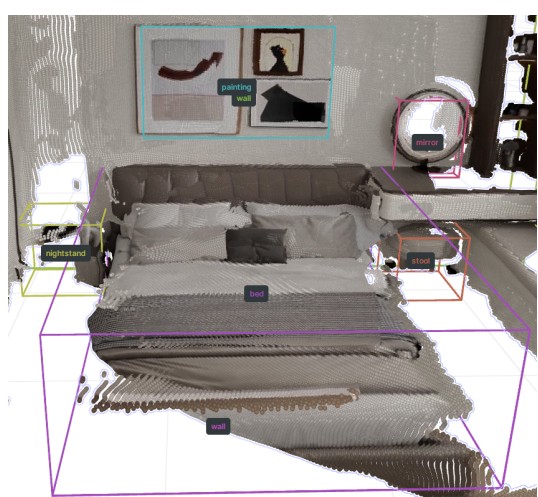

Figure 7: With a 0.8 merge rate, SpatialLM using our method still produces high-quality predictions.

the tokens, the prediction quality remains unaffected. Our method significantly reduces both inference time and memory usage compared to the baseline SpatialLM model, while maintaining comparable or slightly improved detection performance across the evaluated metrics.

## 6 Ablation Study

**Energy Threshold.** We use a threshold $\tau$ to decide which patches $\mathcal{P}$ to aggressively merge. As shown in Table 10 Appendix when $\tau$ is close to $-1$, no patches are merged, resulting in unchanged GFLOPs and a 2% drop in mIoU. As $\tau$ increases, more patches are merged, reducing GFLOPs until they approach the non-adaptive merging baseline. We select $\tau = 0.2$ as it offers the best trade-off, achieving similar GFLOPs to $r = 0.9$ while yielding much higher mIoU.

**Merging Metric** In Table 5, we evaluate the effect of using Q, K, or V features as the merging criterion. We also compare applying the merging function independently per head versus uniformly across all heads. Results show that using the value feature (V) and merging independently per head yields the best performance.

Table 5: Impact of metric and independent head during token matching.

| Metric | Q | K | V |
|--------|---|---|---|
| No Independent Heads | 76.08 | 76.37 | 76.55 |
| With Independent Heads | 76.27 | 76.36 | 76.98 |

Table 6: **OOD-NVS.** Comparisons on the GSO-OOD, Realworld-OOD and Objaverse-OOD evaluation sets with off-the-shelf evaluation. The metric is evaluated on OOD test views with elevation $\phi_{\text{ood}} \geqslant 70\circ$.

| Methods | GSO-OOD | | | Objaverse-OOD | | | RealWorld-OOD | | |
|---|---|---|---|---|---|---|---|---|---|
| | PSNR↑ | SSIM↑ | LPIPS↓ | PSNR↑ | SSIM↑ | LPIPS↓ | PSNR↑ | SSIM↑ | LPIPS↓ |
| MipNeRF360 [3] | 22.90 | 0.824 | 0.192 | 19.6 | 0.72 | 0.28 | 21.99 | 0.878 | 0.127 |
| 3DGS [40] | 21.78 | 0.746 | 0.25 | 19.24 | 0.67 | 0.29 | 23.83 | 0.877 | 0.109 |
| 2DGS [38] | 23.29 | 0.816 | 0.204 | 19.24 | 0.67 | 0.29 | 23.64 | 0.891 | 0.104 |
| Nerfbusters [90] | 15.95 | 0.678 | 0.300 | 16.9 | 0.69 | 0.29 | 23.93 | 0.893 | 0.114 |
| LaRa [11] | - | - | - | 19.0 | 0.68 | 0.32 | - | - | - |
| SplatFormer [14] | 24.71 | 0.857 | 0.152 | 22.43 | 0.808 | 0.179 | 24.33 | 0.900 | 0.100 |
| - Random Drop | 23.77 | 0.821 | 0.19 | 21.80 | 0.777 | 0.208 | 24.02 | 0.889 | 0.105 |
| - Farthest Point S. | 23.29 | 0.817 | 0.194 | 21.13 | 0.757 | 0.223 | 23.91 | 0.889 | 0.107 |
| - VoxelGrid Down. | 23.74 | 0.827 | 0.18 | 21.47 | 0.756 | 0.224 | 23.88 | 0.887 | 0.108 |
| - Our | **24.56** | **0.852** | **0.157** | **22.34** | **0.803** | **0.185** | **24.06** | **0.899** | **0.101** |

# 7 Discussion

Our results show that 3D point cloud transformers rely heavily on excessive tokens, and removing attention layers can simplify the architecture. Using non trainable strided pooling, such as mean pooling, combined with token shuffling for long range information exchange (Figure 8b), achieves competitive performance on ScanNet, with only a slight drop compared to attention based models (76.0 vs 77.0 mIoU). This underscores the importance of attention for long range interactions while motivating more efficient alternatives.

Additionally, applying a 70% token merging rate during downstream training of Sonata preserves performance while significantly reducing computation and training time (Table 7). Unlike standard fine-tuning, where merging is applied after downstream task training, here merging is integrated from the start. This highlights the potential of token merging for efficient training of large models without sacrificing accuracy.

A limitation of our method is that the merging rate $r$ is manually specified rather than learned. Automatically optimizing $r$ under a FLOPs constraint would require an end-to-end framework, but this is challenging because sorting and grouping operations are non-differentiable, necessitating gradient approximations [73, 94, 65]. Another open question is the lack of a formal framework to quantify and reduce redundancy in token representations, which could further improve efficiency and provide stronger theoretical grounding for our approach.

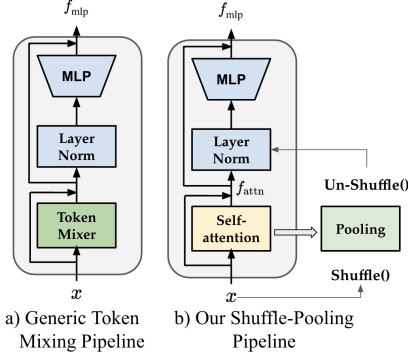

a) Generic Token Mixing Pipeline
b) Our Shuffle-Pooling Pipeline

Figure 8: At each layer, self-attention is substituted with a pooling function, combined with a shuffling function to enable information exchange.

| Version | mIoU | mAcc | allAcc | GPU Mem | GPU Hours |
|---|---|---|---|---|---|
| Sonata-ft | 79.0 | 86.0 | 92.7 | 211.25 GB | 55.2 |
| + Ours | 78.9 | 85.3 | 92.3 | 74.95 GB | 28.3 |

Table 7: Sonata downstream task training performance with and without our token merging method (first row and second row respectively).

# 8 Conclusion

In conclusion, our study reveals that state-of-the-art 3D point cloud transformers are significantly over-tokenized, and their performance can be largely retained even after reducing up to 80-95% of the tokens with a proper merging strategy. We also propose a 3D-specific token merging strategy, integrating local geometric structure and attention saliency to estimate voxel importance, thus enabling aggressive token reduction with minimal accuracy degradation. Our findings not only expose inefficiencies in existing works but also introduce a practical path toward more scalable and computationally efficient 3D vision systems, offering a new perspective on transformer design in 3D tasks and emphasizing the importance of efficient token utilization over parameter scaling.

## Acknowledgement

This work was supported by Deutsche Forschungsgemeinschaft (DFG, German Research Foundation) under Germany's Excellence Strategy - EXC 2075 – 390740016, the DARPA ANSR program under award FA8750-23- 2-0004, the DARPA CODORD program under award HR00112590089. The authors thank the International Max Planck Research School for Intelligent Systems (IMPRS-IS) for supporting Duy M. H. Nguyen. Tuan Anh Tran, Duy M. H. Nguyen, Michael Barz and Daniel Sonntag are also supported by the No-IDLE project (BMFTR, 16IW23002), the MASTER project (EU, 101093079), and the Endowed Chair of Applied Artificial Intelligence, Oldenburg University. Additionally, Hoai-Chau Tran and Khoa D. Doan are supported by the VinUni-Illinois Smart Health Center (VISHC), VinUniversity.

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

# Supplementary Materials for "How Many Tokens Do 3D Point Cloud Transformer Architectures Really Need?"

## Contents

## A   Experiments Setup Details

### A.1   Semantic Segmentation - Datasets and Metrics:

**S3DIS** [2] is a large-scale indoor dataset composed of 3D scans from six areas in office buildings. It includes point-wise semantic annotations across 13 categories, making it a common benchmark for semantic segmentation in indoor environments.

**ScanNet** [16] is a richly annotated dataset of indoor scenes, consisting of RGB-D videos that are reconstructed into 3D meshes. It provides point-wise semantic labels over 20 object categories and is widely used for evaluating 3D semantic segmentation models.

**NuScenes** [8] is an autonomous driving dataset that includes LiDAR point clouds, camera images, and radar data, collected in urban scenes. The 3D semantic segmentation task focuses on labeling LiDAR points across 32 object classes.

**ScanNet200** [98] is an extended version of ScanNet with 200 fine-grained object categories. It introduces a more challenging segmentation task due to its larger label space and long-tail class distribution.

**Metrics:** We evaluate models using several standard metrics. **mIoU** (mean Intersection over Union) measures the average overlap between predicted and ground truth labels across all classes. **mAcc** (mean accuracy) computes the average of per-class accuracies, while **allAcc** (overall accuracy) reflects the proportion of correctly classified points over the entire dataset. In addition to accuracy metrics, we report **FLOPs** (Floating Point Operations) to quantify the computational cost of a model, and **PeakMem** (Peak Memory Usage), which indicates the maximum GPU memory required during inference. These efficiency metrics are critical for understanding model scalability and deployment feasibility.

## A.2 Semantic Segmentation - Baselines:

**ToMe** [5] (Token Merging) is a general framework that reduces token count by merging tokens based on feature similarity, originally proposed for vision transformers. **PiToMe** [81] extends ToMe to 3D point cloud processing by introducing point-wise importance scores to guide the merging process. Both ToMe and PiToMe are limited to merging up to 50% of the tokens.

**ALGM** [66] is a two-stage token merging approach involving global merging followed by local merging. In our adaptation, we use only the local merging stage, which evenly divides tokens into spatial bins and computes intra-bin similarity. Bins containing highly similar tokens are merged based on a similarity threshold. We evaluate three analytic thresholds for merging: $\mu$, $\mu - \sigma^2$, and $\mu - 2\sigma^2$, where $\mu$ is the mean similarity of tokens within a bin and $\sigma^2$ is the variance.

**Point Cloud Downsampling Techniques:** We evaluate several common downsampling strategies for 3D point clouds. **Voxel Downsampling** partitions the 3D space into uniform voxels and retains one representative point per voxel. The feature of each representative point is computed as the mean of the features of all original points within the voxel. **Furthest Point Sampling (FPS)** iteratively selects points such that each newly selected point is as far as possible from previously selected ones, ensuring coverage of the spatial domain. **Random Sampling** simply selects a subset of points uniformly at random from the input set. For all methods, we adjust parameters to ensure that the resulting downsampled point cloud retains approximately 20% of the original points.

# B  Replacements for Attention

Given the significant reduction in tokens achievable during attention computation, a natural question arises: *Is the computationally expensive attention mechanism truly necessary for 3D point cloud processing?* To investigate this, we explore several alternative token mixing strategies by replacing attention with parameter-free or computationally efficient mechanisms (Fig. 9a). The methods are either tested off-the-shelf, fine-tuned (FC) or trained from the scratch (SC). Detailed evaluation is in Tab. 8. The methods evaluated include:

- **ValueFeat**: We eliminate the $O(N^2)$ attention computation involving keys, queries, and values. Instead, we retain only the linear transformation that produces value features. In this setting, we fine-tune the value projection layer and the subsequent MLP layers, while keeping the rest of the network frozen.

- **MLP**: Similar to ValueFeat, self-attention is replaced with a multi-layer perceptron (MLP). However, in this configuration, the entire network is trained from scratch.

- **PoolAttn**: To enhance spatial communication, we first apply average pooling with a large kernel size and stride, both set to $log(N)$ (where $N$ is the number of tokens). Attention is then computed over the pooled features. Finally, unpooling is performed by duplicating the attention outputs back to their corresponding original token positions. is overal attention computation cost is $O(log^2(N))$. This can also be considered a token merging methods that rely purely on locality.

- **AvgPool**: We apply average pooling with a stride of 1 and a large kernel size. Through empirical evaluation, we found that a kernel size of 127 yields the best performance.

- **ShufflePool**: Since AvgPool may lack long-range spatial communication between tokens, we introduce a token shuffling step before pooling. Specifically, we reshape the token sequence as $\text{tokens.reshape}(M, N)$, apply a transpose operation $\text{tokens.swap\_axis}(-1, -2)$, and then flatten it back with $\text{tokens.reshape}(M \times N)$. We then apply average pooling as in AvgPool, using a stride of 1 and a kernel size of 127, which we found to be optimal.

- **Progressive-Tome**: We design an $O(N)$ token merging method by constructing a local graph among tokens, where edges are formed between adjacent tokens in a 1D serialized order. In our experiments, we merge the top 80% of edges with the highest similarity by averaging their corresponding tokens. The un-merging process follows the same approach as our main method.

- **Strided-PiToMe**: We integrate PiToMe with our main approach. Following PiToMe [81], we compute importance scores for each token and select the most important tokens within each bin as the destination set for merging.

**Analysis:** Our results indicate that without long-range spatial information sharing, the model struggles to perform effectively. While the pretrained value features (ValueFeat) retain reasonable performance after fine-tuning (71.3 mIoU), the counterpart trained from scratch (MLP) fails to handle the task adequately, achieving only 42.7 mIoU.

Introducing spatial information sharing through pooling mechanisms (e.g., PoolAttn, AvgPool) leads to significant improvements over ValueFeat (74.8 mIoU vs. 71.3 mIoU), though the communication remains limited to local neighborhoods. To address this, we employ token shuffling followed by pooling and unshuffling. This parameter-free method, trained from scratch, nearly matches the performance of the original attention-based model (76.2 mIoU vs. 77.3 mIoU).

This is a crucial finding: it suggests that attention layers may not be strictly necessary for 3D point cloud processing. Instead, designing effective spatial information sharing mechanisms could offer a more efficient and competitive alternative.

For off-the-shelf token merging methods, we observe that using a metric-based destination selection, such as importance scores from PiToMe, actually degrades performance compared to random sampling. We hypothesize that this is due to the nature of 3D point clouds, where each token corresponds to a single point that is sparse and uninformative on its own. Consequently, importance scores computed at high-resolution layers of the U-Net may not be meaningful or reliable.

Additionally, merging tokens based on pooling, without considering similarity, can negatively impact performance. Pooling tends to indiscriminately combine informative and non-informative tokens, leading to a loss of crucial spatial details.

In contrast, we find that Progressive Token Merging (Prog-Tome), which relies solely on local similarity as the merging criterion, performs comparably to our final method that incorporates both spatial preservation and long-range information sharing (75.5 mIoU vs. 77.0 mIoU). This highlights the effectiveness of localized, similarity-aware merging in point cloud processing.

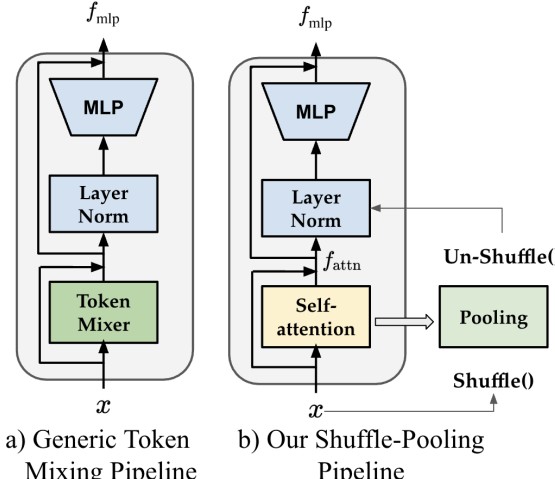

| Methods | mIoU | mAcc | allAcc | Latency (ms) |
|---|---|---|---|---|
| PTv3 (original) | 77.3 | 85.0 | 92.3 | 266 |
| ValueFeat (FT) | 71.3 | 81.1 | 90.6 | – |
| MLP (SC) | 42.7 | 56.2 | 77.4 | – |
| PoolTome (SC) | 42.9 | 57.3 | 72.7 | 194 |
| PoolTome (FT) | 74.8 | 83.0 | 91.2 | 194 |
| AvgPool | 68.6 | 79.1 | 88.9 | – |
| AvgPool (FT) | 74.0 | 81.9 | 91.2 | – |
| ShufflePool (SC) | 76.2 | 84.0 | 92.0 | – |
| Pool-Tome | 71.0 | 81.2 | 90.5 | 194 |
| Prog-Tome | 76.7 | 84.4 | 91.6 | 198 |
| Strided-Pitome | 75.5 | 83.0 | 91.2 | – |
| Ours | 77.0 | 84.5 | 84.5 | 203 |
| Ours (FT) | 77.8 | 92.2 | 92.2 | 203 |

a) Generic Token Mixing Pipeline  b) Our Shuffle-Pooling Pipeline

Figure 9: Visualization of the token mixing module: a) We evaluate different token mixing techniques to replace transformer layers. b) Shuffling the token then perform a pooling layer.

Table 8: Evaluation of attention replacement and token merging methods on ScanNet Val. Methods in the second block are either fully trained from scratch (SC) or fine-tuned (FT). Latency is measured in milliseconds.

## C   Additional Results

### C.1   Segmentation Results

In Table 10, we present detailed results for two configurations: without branching (i.e., merging is performed uniformly across all regions with a fixed merge rate, without using a globally informed graph to selectively merge specific 3D patches) and with branching (i.e., merging is guided by a

Table 9: Dataset statistics and evaluation metrics used in the 3D reconstruction task

| Dataset | Description | Metric | Description |
|---|---|---|---|
| Objaverse [18] | 800K+ (and growing) 3D models with descriptive captions, tags, and animations. | SSIM | Structural Similarity Index measures image similarity based on structure, luminance, and contrast. Higher is better. |
| RealWorld [14] | 4 real-world scenes. | PSNR | Peak Signal-to-Noise Ratio: evaluates reconstruction quality. Higher is better. |
| Google Scanned Objects [23] | More than 1K 3D-scanned household items. | LPIPS | Learned Perceptual Image Patch Similarity: Deep Perceptual Similarity Metric. Lower is better. |

globally informed graph to aggressively combine certain 3D patches). And Figure 10 shows the evaluation of memory consumption during evaluation across different settings.

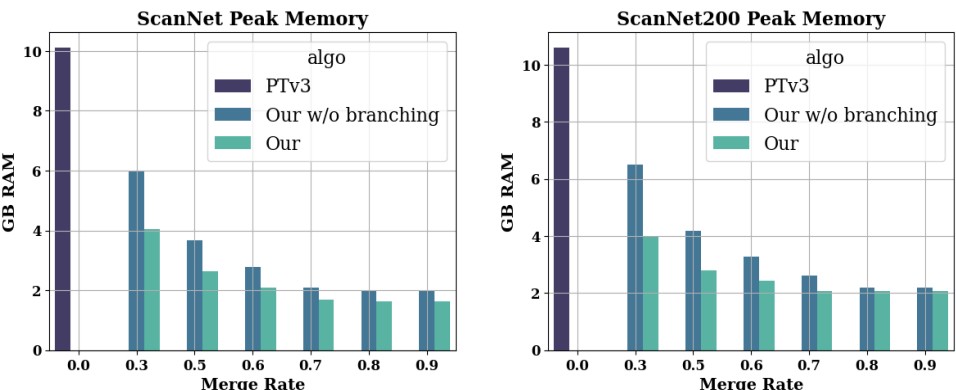

Figure 10: Peak memory consumption evalutation

Table 10: Details results on the segmentation task

| | | ScanNet | | | | ScanNet200 | | | |
|---|---|---|---|---|---|---|---|---|---|
| | | mIoU | mAcc | allAcc | GFLOPS | mIoU | mAcc | allAcc | GFLOPS |
| PTv3 [92] | | 77.68 | 84.77 | 91.82 | 107.5 | 34.57 | 45.58 | 82.79 | 104.99 |
| $+ r = 0.3$ | w branching | 77.60 | 84.40 | 91.79 | 41.37 | 35.10 | 45.03 | 83.20 | 36.40 |
| | w/o branching | 77.63 | 84.62 | 91.91 | 66.98 | 35.09 | 45.58 | 83.29 | 63.89 |
| $+ r = 0.5$ | w branching | 77.62 | 83.91 | 91.57 | 30.48 | 34.72 | 44.37 | 83.06 | 27.79 |
| | w/o branching | 77.69 | 84.59 | 91.80 | 45.73 | 34.76 | 44.92 | 83.16 | 42.65 |
| $+ r = 0.6$ | w branching | 77.45 | 83.71 | 91.48 | 26.43 | 34.48 | 44.07 | 82.96 | 24.62 |
| | w/o branching | 77.51 | 84.55 | 91.79 | 37.80 | 34.52 | 44.55 | 83.09 | 34.72 |
| $+ r = 0.7$ | w branching | 77.20 | 83.53 | 91.39 | 23.32 | 34.21 | 43.74 | 82.90 | 22.17 |
| | w/o branching | 77.31 | 84.60 | 91.81 | 31.63 | 34.29 | 44.25 | 83.02 | 28.54 |
| $+ r = 0.8$ | w branching | 76.98 | 83.41 | 91.34 | 21.10 | 34.20 | 43.70 | 82.98 | 20.42 |
| | w/o branching | 77.11 | 84.22 | 91.81 | 27.17 | 34.24 | 44.13 | 83.06 | 24.09 |
| $+ r = 0.9$ | w branching | 76.24 | 83.05 | 91.36 | 19.75 | 34.38 | 43.64 | 83.21 | 19.39 |
| | w/o branching | 76.40 | 84.17 | 91.80 | 27.14 | 34.54 | 44.13 | 83.28 | 21.44 |

## C.2 Reconstruction Results

We provide additional qualitative results for reconstruction task in Figure 13.

## C.3 Ablation Study on the Number of Bins ($K$)

The central idea of spatial-preserving token merging is to ensure that destination tokens are evenly distributed throughout the input space, which helps maintain both semantic and positional information after merging. In our approach, we aggressively merge all source tokens into destination tokens when the merge rate $r$ exceeds 50%, making it crucial for these destination tokens to be well-dispersed.

Without this binning strategy, multiple tokens could collapse into the same destination token, leading to significant loss of spatial information. By requiring all source tokens within a bin to merge into its corresponding destination token, we promote a more uniform spatial distribution of merged tokens.

To validate this point, we conducted an ablation study on the ScanNet dataset. Table 11 reports the segmentation result (mIoU) for different numbers of bins $K$ and merge rates $r$.

| $r\backslash K$ | 128 | 64 | 32 | 16 |
|---|---|---|---|---|
| 0.5 | 77.04 | 76.67 | 76.78 | 76.80 |
| 0.8 | 76.98 | 76.33 | 75.03 | 73.99 |
| 0.9 | 76.61 | 75.84 | 74.38 | 71.84 |

Table 11: Ablation study on the number of bins ($K$) for different merge rates ($r$). Higher $K$ values generally preserve spatial information better at aggressive merge rates.

These $K$ parameters, however, are not yet optimal. In our approach, we define $K$ dynamically as:

$$K = \lfloor T \cdot (1 - r) \rfloor, \tag{2}$$

where $T$ is the number of tokens in each patch. This formulation ensures that the number of bins adapts naturally to the merge rate, maintaining a balanced spatial distribution of destination tokens.

**Qualitative Comparison: Global vs. Local Energy Score**

We provide a qualitative comparison between global and local energy scores on the ScanNet dataset (without augmentation) to illustrate their impact on performance. Table 12 summarizes the results at different token merge rates.

Table 12: Comparison of global and local energy on ScanNet at various merge rates without test-time augmentation.

| Merge Rate | Global Energy mIoU | Local Energy mIoU |
|---|---|---|
| Without merging | 76.3 | 76.3 |
| $r = 0.3$ | 76.4 | 76.0 |
| $r = 0.5$ | 76.2 | 75.7 |
| $r = 0.8$ | 75.8 | 74.9 |

The results show that global energy scores consistently maintain slightly higher performance than local energy scores, especially at higher merge rates.

# D  Local vs Global energy score

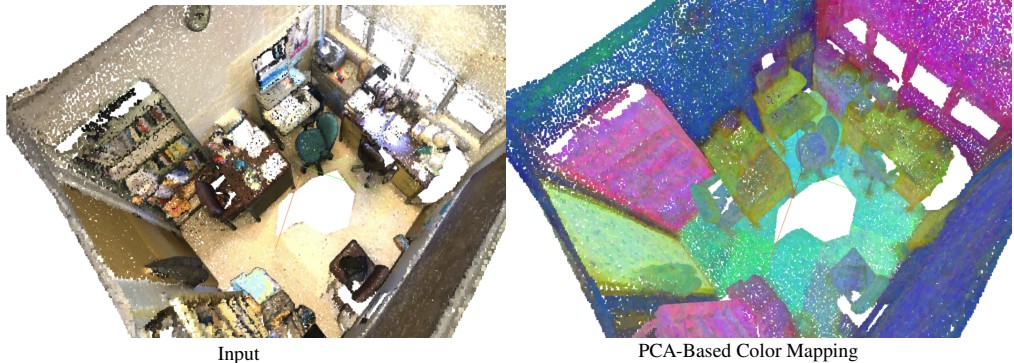

Input        PCA-Based Color Mapping

Figure 11: PCA-Based Color Mapping of all tokens in the last layer of PTv3 model.

To justify the motivation for our globally-informed energy score, we conduct a detailed analysis comparing the behavior of locally-informed energy scores used in PiToMe [81] and explain why it

failed for the 3D Point Cloud models. As demonstrated in Figure 11, most points belonging to the same object exhibit similar features, as indicated by their shared color. This suggests that in the initial and final layers, where each patch's receptive field is still local and covers only a portion of a larger object, individual tokens lack sufficient contextual information. As a result, computing the energy score locally within each patch does not accurately reflect a token's alignment with the global feature space formed by all points in the input point cloud.

To mitigate this limitation, we introduce a globally informed energy score. This involves first computing centroids for each patch, followed by calculating each token's energy score as the average of its alignment with all patch centroids. As illustrated in Figure 12, the globally-informed energy score provides a clearer distinction between foreground and background regions. This enables more effective identification of patches that can be aggressively merged in the initial and final layers. Consequently, tokens representing the foreground are better preserved before entering the middle layers, where the token space is downsampled and each patch has a wider receptive field.

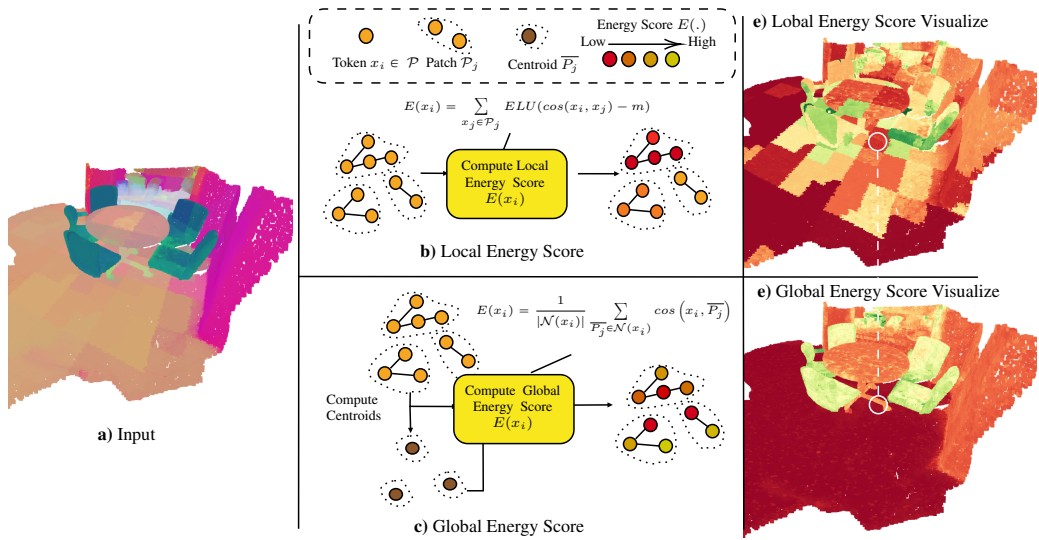

Figure 12: Visualizing Global (Ours) vs. Local (PiToMe[81]) Energy Score for each token

## E    Complexity Analysis

We provide our pseudo code for token merging in Algorithm 1. In our algorithm, the global graph is constructed via matrix multiplication between each point and the patch centroids, resulting in a complexity of $\mathcal{O}(Nkh)$, where $h$ is the dimensionality of the input vectors, $N$ is the number of points, and $k$ is the number of patch centroids (with $k \ll N$). The resulting global energy scores are then used to determine which patches should be aggressively merged.

Let $n$ denote the number of tokens in each aggressively merged patch, and $T$ (where $n \ll T$) be the number of tokens in each patch. The time complexity of the attention operator can be approximated as $\mathcal{O}(k((rT)^2h + n^2h))$ (here $r$ is the ratio of tokens that remain), capturing the dominant contributors to computational cost. However, actual performance may vary depending on the specific PyTorch version and hardware configuration, as optimizations and parallelization may impact the empirical runtime.

## F    Transfer Entropy Analysis for Token Merging

In this section, we take the first step toward formally demonstrating the redundancy and quantifying the amount of information preserved using the Transfer Entropy framework. Following [33, 53], entropy can be used to measure the information content of a network:

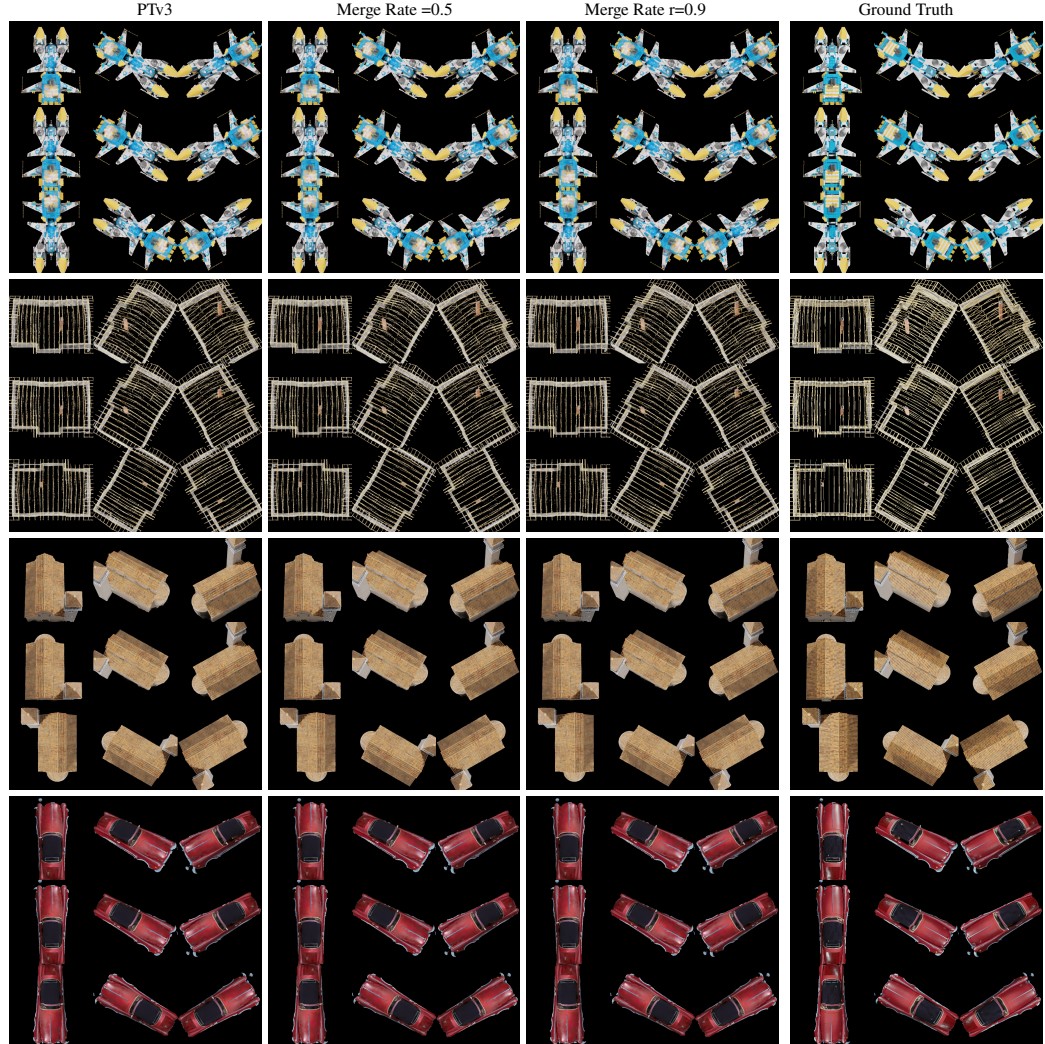

Figure 13: Additional Reconstruction Visualizations Generated by PTv3 with different merge rate

$$H(F) = -\int p(f) \log p(f) \, df, \quad f \in F. \tag{3}$$

Since directly measuring the probability distribution of tokens is non-trivial, we approximate it with a Gaussian distribution [53]:

$$F \sim \mathcal{N}(\mu, \sigma^2). \tag{4}$$

The entropy of a feature set is then:

$$
\begin{aligned}
H(F) &= -\mathbb{E}[\log \mathcal{N}(\mu, \sigma^2)] \\
&= -\mathbb{E}\left[\log\left((2\pi\sigma^2)^{-1/2} \exp\left(-\frac{1}{2\sigma^2}(f - \mu)^2\right)\right)\right] \\
&= \log(\sigma) + \frac{1}{2}\log(2\pi) + \frac{1}{2}.
\end{aligned}
\tag{5}
$$

**Transfer Entropy Definition:** We define Transfer Entropy (TE) as the change in information after applying a token merging function:

## Globally Informed Token Merging

**Input:** Serialized 1D point cloud $\mathcal{P} \in \mathbb{R}^{N \times K \times C}$ (N patches, K points per patch, C features)
**Output:** Merged point cloud representation

### Step 1: Construct Global Bipartite Graph

Compute patch centroids:

$$\bar{P}_j = \frac{1}{|\mathcal{P}_j|} \sum_{x_k \in \mathcal{P}_j} x_k \quad \text{for each patch } \mathcal{P}_j$$

Construct bipartite graph $G = (\mathcal{V}, \mathcal{E})$, where:
- $\mathcal{V} = \{x_i\} \cup \{\bar{P}_j\}$
- $\mathcal{E} = \{(x_i, \bar{P}_j)\}$ — directed edges from points to all patch centroids

### Step 2: Compute Energy Scores

Define outgoing neighbors $\mathcal{N}(x_i) = \{\bar{P}_j \mid (x_i, \bar{P}_j) \in \mathcal{E}\}$
Compute point energy:

$$E(x_i) = -\frac{1}{|\mathcal{N}(x_i)|} \sum_{\bar{P}_j \in \mathcal{N}(x_i)} \cos(x_i, \bar{P}_j)$$

Compute patch energy:

$$E(\mathcal{P}_j) = \frac{1}{|\mathcal{P}_j|} \sum_{x \in \mathcal{P}_j} E(x)$$

### Step 3: Adaptive Merging by Energy

**foreach** *patch $\mathcal{P}_j$* **do**
    **if** $E(\mathcal{P}_j) > \tau$ **then**
        Apply moderate merging $f(\mathcal{P}_j, r)$
    **else**
        Apply aggressive merging $f(\mathcal{P}_j, r^+)$
    **end**
**end**
**return** *Merged point cloud*

**Algorithm 1:** ALGORITHM FOR GLOBALLY INFORMED TOKEN MERGING

$$\text{TE} = H(F) - H(\text{Merged}(F)). \tag{6}$$

This quantifies the amount of information lost or altered due to merging. The Transfer Entropy Rate is defined as:

$$\text{Transfer Rate} = \left| \frac{\text{TE}}{H_{\text{orig}}} \right|. \tag{7}$$

**Experimental Setup:** We analyze the ScanNet validation set with a default merging rate of 70%, reporting transfer entropy rates for different merging scenarios (the $\rightarrow$ indicate the direction of entropy transfer from source to destination):

1. TE A: Original $\rightarrow$ Moderate merging (without Global Informed Graph)
2. TE B: Original $\rightarrow$ Adaptive-aggressive merging (with Global Informed Graph)
3. TE C: Moderate $\rightarrow$ Aggressive merging

| Layer | TE A | TE B | TE C |
|:---:|:---:|:---:|:---:|
| 1 | 0.007 | 0.045 | 0.005 |
| 7 | 0.070 | 0.030 | 0.001 |
| 14 | 0.096 | 0.022 | 0.051 |
| 21 | 0.004 | 0.027 | 0.031 |

Table 13: Layer-wise transfer entropy rates.

**Layer-wise Transfer Entropy Rate Results**

The transfer rates remain consistently small ($< 0.1$) across layers, indicating minimal information loss from compression via merging functions. This aligns with our segmentation performance results and supports the hypothesis, based on a transfer entropy framework, that token representations in point transformer models contain significant redundancy.

While this does not constitute a complete theoretical proof, the transfer entropy framework provides a principled analytical assessment that offers clear evidence supporting our findings. We consider this a starting point for more rigorous theoretical reasoning.

