# OpenReview forum: "How Many Tokens Do 3D Point Cloud Transformer Architectures Really Need?"
_NeurIPS.cc/2025/Conference — NeurIPS 2025 poster_

### Official Review · Reviewer_cv9K · 2025-06-30

**Clarity:** 2
**Significance:** 3
**Originality:** 2
**Rating:** 3
**Confidence:** 4

**Summary:**

In this paper, the authors observe that existing 3D point cloud transformers rely on excessive redundant tokens, yet only a small critical subset is essential for performance. Merging tokens based on spatial importance reduces usage by 90-95% with minimal accuracy loss, enabling efficient scalability across multiple vision tasks.

**Questions:**

(1) Could the authors provide a theoretical analysis or formal justification for the observed token sparsity in their Transformer-based framework? Specifically, what underlying principles or properties of point cloud data lead to this sparsity, and how might this connect to existing theoretical work on token efficiency in attention mechanisms?

(2) Could the authors explicitly clarify the key technical distinctions between their Aggressive Merging Method and the prior approach in [1]?

(3) Explicitly contrast their merging process with RandLA-Net [2], highlighting both algorithmic similarities and any substantively novel aspects?

(4) How does the choice of parameter K in the Figure 4 (c) influence the model’s ability to preserve local geometric structures during the SA Merge operation? Could the authors provide an ablation study analyzing the sensitivity of performance to different K values, particularly given that uniform partitioning may not optimally adapt to varying local point densities?

(5) Could the authors discuss potential challenges or limitations in extending the proposed method to other frameworks beyond PTv3?

[1] Bolya D, Fu C Y, Dai X, et al. Token merging: Your vit but faster[J]. arXiv pre
[2] Hu Q, Yang B, Xie L, et al. Randla-net: Efficient semantic segmentation of large-scale point clouds[C]//Proceedings of the IEEE/CVF conference on computer vision and pattern recognition. 2020: 11108-11117.

**Ethical Concerns:**

["NO or VERY MINOR ethics concerns only"]

**Final Justification:**

I maintain my score, which I believe reflects the strenghts and weaknesses of the work.

**Limitations:**

Yes.

**Paper Formatting Concerns:**

See above.

**Quality:**

3

**Strengths And Weaknesses:**

Strengths
1. Timely and impactful:
- The paper tackles a highly relevant issue—scalability and inefficiency in transformer-based 3D models.
- It challenges the implicit assumption that “more tokens = better performance,” backed by robust empirical evidence.

2. Methodological novelty:
- The proposed global-informed energy score for adaptive merging is well motivated and exploits domain-specific spatial structure, improving upon generic token reduction strategies.

3. Strong empirical validation:
- Experiments span three core 3D tasks with multiple datasets (ScanNet, S3DIS, ShapeNet, Objaverse, GSO, Waymo), showing generality.
- Ablations on merging strategies, thresholds, and feature types (Q, K, V) are interesting.

4.	Practical implications:
- The method operates inference-time without retraining, and can even outperform the baseline with minimal fine-tuning.
- Clear gains in FLOPs and memory make the work valuable for real-world deployment (e.g., mobile robotics, AR/VR).

5.	Comprehensive related work and discussion:
- Covers point cloud, transformer compression, and downsampling literature in a balanced way.
- Acknowledges open problems (e.g., automatic merging rate optimization, extension to meshes or real-time streams).

Weaknesses:
- While this work is primarily motivated by empirical observations, it lacks a rigorous theoretical analysis regarding the inherent sparsity of tokens in Transformer-based architectures.

- The proposed Aggressive Merging Method demonstrates only marginal technical differentiation from the prior approach [1], thereby constraining the innovative contribution of this work.

- The methodological pipeline presented in Figure 4 lacks sufficient detail to clearly elucidate the functional relationships between components (a), (b), and (c). Furthermore, the figure contains labeling inaccuracies, particularly in the misassignment of r⁺ and r in (a), and the incorrect color coding of source (red) and destination (blue) points in (c). Upon closer examination, the merging process appears fundamentally similar to the down-sampling strategy employed in RandLA-Net [2], raising questions about the claimed technical novelty of this aspect of the proposed method.

- The manuscript fails to provide a formal definition for term ‘E’ introduced in line 191, which impedes proper understanding of the subsequent mathematical derivation.

- The ablation study fails to evaluate the impact of parameter K in Figure 4 (c). The current approach of uniformly dividing each patch into K spatial bins may adversely affect the model’s ability to capture local geometric structures in the SA Merge operation.

- While I greatly appreciate the thorough experiments and validations presented in the supplementary materials, the proposed method has only been validated on the PTv3 framework. This limited scope makes it difficult to assess the generalizability of the approach. To strengthen the claims, the authors should demonstrate its effectiveness across additional baselines or diverse benchmarks.

[1] Bolya D, Fu C Y, Dai X, et al. Token merging: Your vit but faster[J]. arXiv pre
[2] Hu Q, Yang B, Xie L, et al. Randla-net: Efficient semantic segmentation of large-scale point clouds[C]//Proceedings of the IEEE/CVF conference on computer vision and pattern recognition. 2020: 11108-11117.

---

> ### Author Rebuttal · Authors · 2025-07-31
>
> Thank you for the insightful feedback, we would like to answer your questions as follows:
> ## Why tokens in point cloud transformers are highly redundant?
>
> Thank you for the question, it touches on a key motivation behind our work and points to an important direction for future research. Our intuition regarding the high token redundancy in point cloud transformers stems from the inherent properties of 3D data similar to how single pixels in a 2D image are not very informative on their own. Unlike in Vision Transformers, where semantic information is often captured in structured patches (e.g., 16×16 pixels), 3D point clouds are unordered, spatially noisy, and frequently incomplete. Individual points in 3D space typically carry very limited information, and many of them correspond to repetitive structures within larger objects, such as flat surfaces or repeated geometric patterns, which further amplifies redundancy.
>
> This perspective is further supported by the design of state-of-the-art methods like Point Transformer v3 and Sonata, which rely on fine-grained attention mechanisms to model local context. In both the first and last layers of these transformer backbones, point tokens tend to exhibit high redundancy, a phenomenon we visualize using PCA projections. At these stages, individual tokens contribute minimal unique information, making self-attention computationally expensive and inefficient. Nevertheless, it's important to retain a subset of these tokens, as some may belong to semantically rich or structurally complex objects that become meaningful only when captured by the broader receptive fields of deeper layers. Identifying which tokens to preserve is non-trivial, and this challenge motivates our proposed globally-informed graph representation, which guides token selection based on global context.
>
> In future work, we plan to explore this redundancy more formally by applying the transfer entropy framework to analyze and justify the globally-informed graph representation introduced in our method. We believe this will offer both theoretical insights and practical improvements for designing more efficient transformer architectures in the 3D domain.
>
> ##  Aggressive Merging Method versus ToMe
> - Findings: ToMe and other prior works on Vision Transformers primarily focus on 2D images, where each token typically covers a relatively large and semantically meaningful region of the image. This makes individual tokens highly informative, which in turn limits the degree of safe token merging, usually only a small fraction of tokens are merged progressively at each layer to avoid significant information loss. In contrast, our study highlights a key difference in the 3D domain: transformer backbones such as PTv2, PTv3, and Sonata frequently encounter substantial token redundancy. This might stems from the structural repetition inherent in 3D scenes. As a result,  we observe that the attention mechanism can become overused on redundant or uninformative tokens, particularly in the early and late layers where token dimensionality is shallow. We believe these findings will open up new directions for designing more efficient transformer architectures tailored to the unique characteristics of 3D data.
>
> - Method: Leverage our finding, we propose a token compression strategy that aggressively reduces redundancy by identifying and merging tokens in low-energy regions of the 3D scene. Our method employs a globally-informed energy scoring mechanism to quantify and detect  redundant regions. By merging up to 99% of the tokens, while preserving the most informative ones, we demonstrate that a well-design token compression method can significantly reduces computational cost with negligible impact on accuracy.
>
>
> ## Ablation study for the number of bins ( $K$ )
> The central idea of spatial-preserving token merging is to ensure that destination tokens are evenly distributed throughout the input space, which helps maintain both semantic and positional information after merging. In our approach, we aggressively merge all source tokens into destination tokens when the merge rate $r$ exceeds 50%, making it crucial for these destination tokens to be well-dispersed. Without this binning strategy, multiple tokens could collapse into the same destination token, leading to significant loss of spatial information. By requiring all source tokens within a bin to merge into its corresponding destination token, we promote a more uniform spatial distribution of merged tokens. To validate this point, we conducted an ablation study on the ScanNet dataset.
>
>
> |               | $k=128$| $k=64$ | $k=32$| $k = 16$ |
> |----------|------- |------|------|----|
> | $r=0.5$    | 77.04  | 76.67 |76.78| 76.80|
> | $r=0.8$    | 76.98  | 76.33| 75.03| 73.99|
> | $r=0.9$    | 76.61  | 75.84| 74.38| 71.84|
>
> These $K$ parameters however is not yet optimal, in our paper we define $k$ as: $\lfloor T * (1 - r)\rfloor$ , where $T$ is the number of token in each patch.
>
> ## Comparision with  RandLA-Net
>
> - A key difference between RandLA-Net and our method lies in their design philosophy and integration. Our method is an off-the-shelf technique aimed at reducing redundant information and computational overhead inherent in large-scale point transformer architectures such as PTv3 and the foundation model Sonata. In contrast, RandLA-Net is an independent architecture that must be trained from scratch.
> - Our framework also includes an analytical tool that identifies redundant regions in the point cloud using a proposed globally-informed graph structure, enabling more effective compression. Before comparing our method directly with RandLA-Net, it is instructive to contrast the underlying architectures of PTv3/Sonata and RandLA-Net.
> While both PTv3 and RandLA-Net are designed for large-scale point clouds (on the order of millions of points), they adopt fundamentally different strategies. RandLA-Net employs random sampling and local feature aggregation based on KNN, which can become computationally expensive and less scalable. PTv3, however, demonstrated that the KNN-based aggregation used in RandLA-Net imposes a computational bottleneck and limited receptive field. To address this, PTv3 introduced a serialization scheme that orders and groups points sequentially, replacing the costly KNN approach.
> - Our method builds upon PTv3's serialization strategy by dividing space into smaller scales and selecting a representative set of DST tokens, rather than sampling points randomly as RandLA-Net does. This preserves spatial structure more effectively and avoids the clustering issues inherent to random sampling. Moreover, our method performs aggregation using DST tokens at a more global scale (e.g., 1000 points), as opposed to RandLA-Net's local KNN-based selection, further enhancing efficiency and representation power.
>
> ## Beyond PTv3 and Sonata
> PTv3 and Sonata are the first large-scale, scene-level point cloud transformers capable of performing self-attention across thousands of points simultaneously. This design allows them to capture fine-grained and important details during training. However, it also leads to significant redundancy in the data, an issue our method aims to address.
>
> In contrast, other techniques mitigate computational cost by partitioning the point cloud before processing, either through spatial grouping (e.g., RandLA-Net, Yogo [1], SuperPoint [2]) or via 1D serialization with localized self-attention (e.g., OctFormer [3], which attends to up to 128 points). The grouping-based methods typically exhibit lower redundancy due to their coarse representation and small number of groups, meaning that pruning one or two groups has limited impact on efficiency. Meanwhile, in serialization-based approaches, the self-attention layer is not the main bottleneck, the point cloud itself is. Thus, a different strategy is needed, such as completely or temporarily pruning redundant points.
>
>
> Given the demonstrated strength of PTv3 and Sonata as state-of-the-art 3D architectures and foundational models for 3D pretraining, we anticipate that future architectures will build upon them. We believe integrating our redundancy optimization method into the training process of such models is a promising direction, particularly as our work so far has focused primarily on improving inference efficiency.
>
>
> [1] You Only Group Once: Efficient Point-Cloud Processing with Token Representation and Relation Inference Module
>
> [2] Efficient 3D Semantic Segmentation with Superpoint Transformer
>
> [3] Octformer: Octree-based transformers for 3d point clouds

---

> > ### Comment · Reviewer_cv9K · 2025-08-06
> >
> > Thank you for the detailed and thoughtful rebuttal. I appreciate the authors’ comprehensive effort in addressing the concerns raised in the review. Below I provide specific feedback on the points discussed in the rebuttal:
> >
> > 1. On Token Redundancy in 3D Point Cloud Transformers
> > The expanded explanation regarding the inherent properties of 3D data and its contribution to token redundancy is helpful and well-articulated. The connection to PCA visualizations and comparisons with structured 2D inputs strengthens the intuition. However, the rebuttal still lacks a formal theoretical grounding or analytical justification—which was the crux of the original question. Future work in this direction, as mentioned, would indeed be welcome.
> >
> > 2. Aggressive Merging vs. ToMe
> > The clarification is useful, particularly in articulating how redundancy manifests differently in 3D data compared to 2D, and how your globally-informed energy scoring offers a more tailored solution. Still, the technical distinction could be reinforced with more empirical comparisons beyond the conceptual differentiation—for example, comparing against ToMe variants adapted for point cloud settings (if feasible).
> >
> > 3. Ablation Study on Parameter K
> > Thank you for adding the ablation results on the number of bins (K). These results are informative and help demonstrate the robustness of your method under different bin sizes. However, this analysis remains limited to a single dataset (ScanNet), and it is still unclear whether the uniform binning strategy would generalize across datasets with highly varying local point densities (e.g., outdoor scenes). A discussion of this limitation or future direction would be helpful in the paper.
> >
> > 4. Comparison with RandLA-Net
> > The architectural distinctions are now better highlighted, especially regarding the serialization strategy versus local KNN sampling. However, in Figure 4. c), the process is very similar to the key component of RandLA-Net. The random selection is used here. In my opinion, the proposed work is the combination of PTv3 and RandLA-Net. In this rebuttal, the authors did not clarify the essential differences between the proposed method and previous approaches (PTv3 and RandLA-Net).
> >
> > 5. Beyond PTV3 and Sonata
> > It’s appreciated that the authors acknowledge the limited scope of evaluation and express intent to explore generalizability beyond PTV3/Sonata. However, this remains a major limitation. Demonstrating efficacy on other frameworks—or at least discussing expected challenges—would significantly strengthen the work's broader impact claims.
> >
> > Conclusion
> > The rebuttal strengthens the case for the paper in several ways, particularly in clarifying motivations, offering preliminary ablation results, and contrasting with related methods. Nevertheless, the original concerns—particularly regarding theoretical justification, novelty relative to existing methods (e.g., RandLA-Net), and limited empirical generalization—still stand to a large extent.
> >
> > Therefore, while the rebuttal adds clarity and value, it does not, in my view, warrant a change in the overall assessment.

---

> ### Author Response · Authors · 2025-08-08
> **Response to Reviewer cv9K - Follow up**
>
> We thank the Reviewer for the follow-up concerns and the opportunity to clarify our contributions. We would like to emphasize the following key points. Below we address your remaining concerns.
>
> **1. Requiring Formal Theoretical Results**
>
> We would like to emphasize the following key points:
>
> **a. Novel Empirical Discovery for Efficient 3D Point Cloud Transformer:**
>
> Our paper is the **first to systematically uncover and quantify token redundancy in large-scale 3D point cloud transformer architectures, including PTv3 and the foundation model Sonata**. This insight is crucial for the community, as it exposes a significant inefficiency in current state-of-the-art models, models that are increasingly used despite their heavy computational and energy requirements. We believe that findings will draw the community’s attention to real-world deployment of these architectures, **which incur substantial and avoidable costs, particularly in resource-constrained environments**.
>
> **b. Comprehensive Experimental Validation:**
>
> We conduct an extensive empirical study involving a range of existing token reduction and merging techniques, alongside our own novel method that captures spatial characteristics specific to 3D point clouds. These methods are evaluated across multiple architectures (PTv3, Sonata) and diverse tasks, including indoor and outdoor segmentation, and point cloud reconstruction. This broad validation underscores the robustness and generalizability of our findings.
>
> **c. Impactful Contributions and Future Directions:**
>
> While we appreciate the Reviewer’s interest in theoretical analysis, we respectfully argue that the strength of our paper lies in its empirical rigor and practical insight. **Our contributions lay a foundation for a number of impactful future research directions**, including:
>
> - Investigating whether token redundancy is prevalent in other non-transformer 3D point cloud architectures.
>
> - Designing new transformer variants specifically optimized for token efficiency in 3D vision tasks.
>
> - Developing task-specific token merging or compression strategies for further performance gains.
>
> - Theoretical studies to understand the underlying reasons behind token redundancy in 3D models.
>
> We believe that **requiring a formal theoretical proof - while valuable - should not be viewed as a prerequisite** for validating the novelty or significance of our contributions. Rather, **we see theoretical understanding as a natural and important future step**, which our work motivates and opens the door for.

---

> ### Author Response · Authors · 2025-08-08
> **Response to Reviewer - Continued (2)**
>
> **2. Theoretical Directions to Analyze Token Redundancy in 3D Point Cloud Transformers**
>
> While our paper only provides some intuition as a previous response, we also present here a few more concrete directions toward a feasible theoretical foundation. In particular, one promising idea is to **construct a token interaction graph** where each node represents a token (e.g., a voxel or point embedding), with edges weighted by either self-attention affinities or spatial proximity. This graph can serve as a structured representation for deeper theoretical study.
>
> **a. Spectral Graph Coarsening as a Framework for Token Merging Evaluation**
>
> Inspired by PiToMe [1], which employs spectral graph theory to preserve intrinsic spectral properties when merging tokens, one can adapt this framework to 3D point cloud transformers. Specifically, consider using the token connectivity graph and perform graph coarsening by merging nodes with similar local neighborhoods while ensuring that the spectrum (i.e., eigenvalues of the normalized Laplacian) between the original and the coarsened graph remains close.
>
> **b. Topological Redundancy via Persistent Homology**
>
> We can then further analyze **which tokens are topologically essential by using persistent homology** to extract features like connected components, loops, and cavities from the point cloud or the token graph. Then, test whether removing certain tokens alters the topological signatures significantly. Tokens whose removal leaves those signatures mostly unchanged can be considered topologically redundant, a theoretical grounding for merging them without harming downstream task performance.
>
> **c. Unified Spectral–Topological Framework**
>
> By merging the above ideas, a joint framework can be:
>
> - **Graph Spectral Stability**: Use **spectral distance metrics** to quantitatively gauge how similar a coarsened (merged) token graph is to the original in terms of eigenvalue distributions. A small spectral distance indicates that the global relational structure is maintained.
>
>
>
>
> - **Topological Invariance**: At the same time, **one can measure how topological invariants, such as Betti numbers [2,3]**, which capture essential geometric features like connected components and holes, change when tokens are removed or merged. If these invariants remain stable, it suggests that the overall structure of the point cloud is preserved despite the reduction in tokens.
>
> Such an approach has the potential to deliver a rigorous theoretical basis showing that merging redundant tokens can substantially reduce computational cost, while safeguarding both structural and topological integrity of the data representation.
>
> [1] Accelerating Transformers with Spectrum-Preserving Token Merging, NeurIPS 2024
>
> [2] Wang, Hao. "Betti Number for Point Sets." Journal of Physics: Conference Series. Vol. 2555. No. 1. IOP Publishing, 2023.
>
> [3] Paul, Rahul, and Stephan Chalup. "Estimating Betti numbers using deep learning." 2019 International Joint Conference on Neural Networks (IJCNN). IEEE, 2019.
>
> **3. Unclear if the proposed method remains robust to outdoor scenes**
>
> We thank the Reviewer for the thoughtful and detailed feedback. **We would like to clarify that results for outdoor scenes were already included in the supplementary material**; however, we apologize if this was not sufficiently highlighted or easy to locate. To address this, we provide a concise summary of our findings for this setting below:
>
> - We ran experiments with **LiDAR point clouds** on off-the-shelf results for the **nuScenes dataset** [2], presented in Table 7 of the supplementary material. Specifically, we report results using a standard merge rate of 80% and an aggressive merge rate of 95% (K=32). A summary is provided below:
>
> | Method        | mIoU | mAcc | allAcc | peakMem (GB) | GFLOPS | Latency (ms) |
> |-------|----|------|---|-----|----|---|
> | PTV3          | 80.3 | 87.2 | 94.6   | 6.20  | 101.68 | 152  |
> | PTV3 + Ours   | 78.0 | 85.5 | 94.0   | 0.92   | 32.45  | 106 |
>
> The results suggest that our binning strategy generalizes well to outdoor point clouds with highly variable local point densities.
>
> Additionally, our method also works with object-centric datasets, including
>
> - **For the 3D reconstruction task** with ObjectVerse and Google Scanned Object datasets, please see Sec. 5.3, Fig. 6, and Fig. 7 in our main manuscript.
> - **For the 3D part segmentation** with SAMPart3D[1] model (conducted during rebuttal).
>
> | Method            | Human-Shape | Animals | Daily-Used | Buildings&Outdoor | Transportations | Plants | Food  | Electronics | Total Mean AP | Peak GPU Mem |
> |-----|-------|---------|-----|---|-----|-----|-------|---|-------|-----|
> | SamPart3D         | 0.4454      | 0.4224  | 0.3911     | 0.3109  | 0.3683  | 0.4466 | 0.4693 | 0.3576       | 0.3904 | 3.44GB         |
> | SamPart3D + Ours  | 0.4182      | 0.4194  | 0.3782     | 0.2895 | 0.3329  | 0.4113 | 0.4670 | 0.3955       | 0.3785 | 1.45GB         |

---

> ### Author Response · Authors · 2025-08-08
> **Response to Reviewer - Continued (3)**
>
> In which we applied a default merge rate of 0.6 and an aggressive merge rate of 0.9 for the experiments. We did not have enough time to complete our analysis for the GFLOPs in these experiments. However, we reported a significant drop of memory consumption from 3.44GB to 1.45GB while only having 1.2% drop in total AP score. This suggests that we would likely observe a huge drop rate with GFLOPs. Fine-tuning experiment on this dataset is also promising.
>
> [1] Yang, Yunhan, et al. "Sampart3d: Segment any part in 3d objects." arXiv preprint arXiv:2411.07184 (2024).
>
>
> **4. Clarify the essential differences between the proposed method vs (Ptv-3 and RandLA-Net).**
>
> We thank the Reviewer for pointing out a potential confusion with RandLA-Net, and we appreciate the opportunity to clarify this.
>
> While **RandLA-Net** is a **standalone network architecture** designed for point cloud processing (notably following a U-Net style structure), our method is a **token merging strategy designed to be integrated into existing point transformer** architectures (e.g., PTv3, Sonata) to reduce redundancy in a principled and efficient manner.
>
> **A. Key Differences:**
> RandLA-Net uses:
>
> - **Random sampling** for point cloud downsampling, which may result in loss of spatial structure and does not consider feature-level redundancy.
>
> - **Local feature aggregation** (LFA) modules using KNN (K=16) with small receptive fields, limiting global context awareness.
>
> - **Nearest neighbor interpolation** during decoding, which can introduce approximation errors in dense reconstructions.
>
>
> In contrast, **our method**:
> - **Is not a downsampling or point reduction technique**, but rather a token merging mechanism guided by spatial and feature similarity.
>
> - Performs **bin-wise selection of representative tokens** and computes token redundancy across **multiple self-attention heads**, preserving spatial fidelity and feature diversity.
>
> - Utilizes a **large receptive field (≥1000 points)** for redundancy detection and performs recovery via **head-wise recomposition, not interpolation**.
>
> **B. Additional Comparison with RandLA-Net-Style Pipeline**
>
> To further validate the differences, we conducted an **adapted RandLA-Net implementation within PTv3 for a direct comparison**. In this baseline:
>
> - We apply random sampling, followed by KNN feature pooling (K=16),
>
> - Run self-attention on the pooled set,
>
> - And use nearest neighbor interpolation to recover the original token positions, mimicking the RandLA decoding.
>
> As shown in the table below (see attached figure), our method outperforms this baseline **consistently across multiple reduction ratios (r)**. This highlights the advantages of our approach in terms of **spatial structure preservation, head-wise processing, and feature-guided compression**.
>
> | Method  | r=0.8 | r=0.9 | r=0.95 | r=0.99 |
> |-------|-------|-------|--------|---|
> | PTv3 + RandLA-Net | 74.90 | 72.07 | 67.66  | 46.84  |
> | PTv3 + Ours   | **77.03** | **76.40** | **75.51**  | **71.25**  |
>
> In summary, we emphasize that:
> - RandLA-Net is a **specific architecture** with a fixed sampling and aggregation mechanism;
>
> - Our method is a **general, plug-and-play token merging** strategy applicable across architectures;
>
> - **The superiority of our method** is empirically demonstrated through a carefully controlled comparative experiment.
>
> We hope this clears up any confusion and reinforces the novelty and generality of our proposed approach.
>
> **5. Applying in another framework outside PTV3/Sonata**
>
> We thank the Reviewer for the insightful suggestion. As highlighted in our contributions, **our primary objective was to uncover and address token redundancy in large-scale point cloud transformer architectures**, specifically in PTv3 and Sonata—the most recent and relevant state-of-the-art models designed for dense point cloud representations with **large token counts (≥1000 tokens)** and wide receptive fields.
>
> Our method is particularly well-suited to such settings, where redundancy becomes computationally costly. In contrast, earlier architectures often operate with significantly fewer tokens, where redundancy is minimal and the gains from merging would be negligible. This explains our focus on PTv3 and Sonata, where the impact of our method is measurable and meaningful.
>
> We agree that integrating our method into other architectures is a valuable direction. For instance, **RandLA-Net could theoretically adopt our merging approach after each Local Spatial Encoding layer**. However, since it relies on local aggregation with small receptive fields and relative positional encodings, merging may disrupt its core operation, requiring non-trivial adaptations.
>
> In summary, while our current focus is on dense models, our method remains architecture-agnostic and well-suited for broader adoption as future architectures move toward larger receptive fields and denser inputs.

---

> ### Author Response · Authors · 2025-08-08
> **Response to Reviewer - Continued (4)**
>
> **6. Comparing against merging as ToMe designed for 3D point cloud**
>
> We conduct an extensive empirical study involving a range of existing token reduction and merging strategies, including adaptations of principal mechanisms used in ToMe (Token Merging) and related approaches. While ToMe has shown success in 2D vision tasks, **we found no existing implementation or variant tailored for 3D point cloud transformers. This is likely because, to the best of our knowledge, our work is the first to reveal and systematically study token redundancy in large-scale 3D point cloud architectures**.
>
> To enable a fair and meaningful comparison, we adapted ToMe’s core idea of merging based on feature similarity to the 3D domain and evaluated it alongside our own method, which is specifically designed to preserve spatial structures unique to point clouds. All methods were rigorously tested across multiple strong transformer architectures (PTv3 and Sonata) and diverse downstream tasks, including indoor and outdoor semantic segmentation, and point cloud reconstruction.
>
> This broad and thorough evaluation not only highlights the robustness and generality of our findings, but also emphasizes the distinct advantage of our approach, which accounts for the spatial and structural complexities of 3D data—something generic 2D-based techniques like ToMe are not equipped to handle out of the box.
>
> In summary, **we see this work as a first step toward a larger research direction**. By identifying and demonstrating token redundancy in 3D transformers, we hope to inspire the community to further explore and improve token merging mechanisms tailored for 3D data. **Our method provides an initial, effective solutions, yet we believe more advanced and higher-performing approaches will naturally follow as this research direction gains traction**.

---

> ### Author Response · Authors · 2025-08-09
> **Response to Reveiwer - Continued (5)**
>
> # Transfer Entropy Analysis for Token Merging
>
> Following [1, 2], entropy can be used to measure the information quantity of a network:
>
> $$
> H(F) = - \int p(f) \log p(f) \, df, \quad f \in F.
> $$
>
> Since directly measuring the probability distribution of tokens is non-trivial, we approximate it with a Gaussian distribution [2]:
>
> $$
> F \sim \mathcal{N}(\mu, \sigma^2).
> $$
>
> The entropy of a feature set is then:
>
> $$
> H(F) = -\mathbb{E}[\log \mathcal{N}(\mu, \sigma^2)] \\
> = -\mathbb{E}\left[\log\left((2\pi\sigma^2)^{-1/2} \exp\left( -\frac{1}{2\sigma^2}(f-\mu)^2 \right)\right)\right]  \\
> = \log(\sigma) + \tfrac12 \log(2\pi) + \tfrac12.
> $$
>
> ---
>
> ## Transfer Entropy (TE) Definition
>
> We define **Transfer Entropy** as the change in information after applying a token merging function:
>
> $$
> \mathrm{TE} = H(F) - H(\mathrm{Merged}(F)).
> $$
>
> This quantifies the amount of information lost or altered due to merging.
>
> The **Transfer Entropy Rate** is:
>
> $$
> \text{Transfer Rate} = \left| \frac{\mathrm{TE}}{H_{\mathrm{orig}}} \right|.
> $$
>
> ---
>
> ## Experimental Setup
>
> We analyze the ScanNet validation set with a default merging rate of 70%, reporting transfer entropy rates for:
>
> - **TE A**: Original → Moderate merging (w/o Global Informed Graph)
> - **TE B**: Original → Adaptive-aggressive merging (w/ Global Informed Graph)
> - **TE C**: Moderate → Aggressive merging
>
> ---
>
> ## Layer-wise Transfer Energy Rate Results
>
> | Layer | TE A   | TE B   | TE C   |
> |-------:|:------:|:------:|:------:|
> | 1     | 0.007  | 0.045  | 0.005  |
> | 7     | 0.070  | 0.030  | 0.001  |
> | 14    | 0.096  | 0.022  | 0.051  |
> | 21    | 0.004  | 0.027  | 0.031  |
>
> ---
>
> **Observation:** Transfer rates remain consistently small (< 0.1) across layers, indicating minimal information loss from compression via merging functions. This aligns with our segmentation performance results and supports the hypothesis—based on a transfer entropy framework—that token representations in point transformer models contain significant redundancy.
>
> While this does not constitute a complete theoretical proof, the transfer entropy framework provides a principled analytical assessment that offers clear evidence supporting our finding. We believe this is a good starting point for a more rigourous theoretical reasoning.
>
> [1] Guan, Chaoyu, et al. "Towards a deep and unified understanding of deep neural models in nlp." International conference on machine learning. PMLR, 2019.
> [2] Lin, Sihao, et al. "Mlp can be a good transformer learner." Proceedings of the IEEE/CVF Conference on Computer Vision and Pattern Recognition. 2024.

---

### Official Review · Reviewer_4bMr · 2025-06-30

**Clarity:** 4
**Significance:** 3
**Originality:** 3
**Rating:** 4
**Confidence:** 3

**Summary:**

The paper puts forward an interesting point of view: only 5%-10% of the tokens are needed to represent the original complex and redundant token sequence without significantly affecting the performance. Based on this idea, the paper proposes a specific token merging strategy for 3D transformer, which integrates local geometric structure with attention saliency to evaluate the importance of voxels, thereby achieving significant token reduction while maintaining minimal accuracy loss. Finally, a large number of experiments verify the effectiveness of the proposed token merging strategy in 3D semantic segmentation, reconstruction and detection tasks. It is worth mentioning that the proposed method can even surpass the baseline through a fine-tuning strategy, after removing 90%-95% of the tokens.

**Questions:**

In addition to the weaknesses mentioned, please solve the following confusions:
1. Can other token merging strategies, such as ToMe, ALGM, PiToMe, etc., improve their compression rate so that GFLOPS remains at a level comparable to that in this article, and then evaluate their performance? For example, Figure 2.
2. In Table 1, why did the performance decrease after fine-tuning on the S3DIS dataset? This result should be discussed specifically.

**Ethical Concerns:**

["NO or VERY MINOR ethics concerns only"]

**Limitations:**

Yes

**Quality:**

3

**Strengths And Weaknesses:**

**Strengths**:
1. This paper proposes a specific token merging strategy for 3D transformer, which evaluates the importance of voxels by integrating local geometric structure with attention saliency, thereby achieving significant token reduction while maintaining minimal accuracy loss.
2. This paper conducts a large number of experiments to verify the effectiveness of the proposed token merging strategy in 3D semantic segmentation, reconstruction and detection tasks. It is worth mentioning that the proposed method can even surpass the baseline by fine-tuning the strategy while removing 90%-95% of the tokens.

**Weaknesses**:
1. The method is too simple. The proposed token merging method does not involve any parameter updates, and the method is too simple.
2. Lack of more experimental verification. The method proposed in this paper should be plug-and-play. In addition to PTv3, it can also be adapted to other transformer-based models in theory. The author should do more experiments to verify the feasibility and applicability of this method.
3. The icon title and caption are wrong. Table 2 should actually be Figure 6.

---

> ### Author Rebuttal · Authors · 2025-07-30
>
> Thank you for the insightful feedback. We will incorporate both your concerns and the discussion into our manuscript.
> ## Other Model
>
> Theoretically, our technique can be applied to other point transformer models such as **OctFormer**[2] or **FlatFormer**[1]. However, **PTv3** and **Sonata** are the first transformer-based models to operate on point clouds at a fine-grained level, processing self-attention across more than 1000 points simultaneously. This scalability is particularly beneficial for large-scale scenes, where millions of points may exist with uneven density and varied semantic importance (e.g., a few points may hold crucial information while many do not). This makes point-level self-attention crucial.
>
> In contrast, other models group points first (e.g., **PointNet**, **PointNet++**), then apply attention with a limited number of points (up to 128 for **OctFormer** and **FlatFormer**), limiting their capability. This also reduces attention overhead, but at the cost of representation power.
>
> Notably, **Sonata** is the first pretrained foundation model at the scene level, with a large receptive field for self-attention. It has also been used to power 3D-language models such as **SpatialLM**[3].
>
> ## Simplicity
>
> A key insight in our work is that point transformer models overuse tokens, leading to excessive computation and memory usage. Our globally-informed graph provides additional context to the local transformer, helping determine which areas contain more informative tokens.
>
> The simplicity of our method ensures robustness and can be used to for future design decisions in efficient 3D point cloud processing.
>
> ## Adaptation of Other Methods
>
> To extend the evaluation scope for other baselines, such as ALGM, we tested their performance across different threshold settings. For PiToMe, applying merging rates above 50% is theoretically infeasible due to its formulation, which requires maintaining a set of preserved tokens. When the merging ratio exceeds 50%, all tokens become candidates for merging, resulting in the concept of a "preserved set" invalid.
> We explored adapting their energy-based formulation to determine a set of protected tokens, but found that this approach yielded worse results compared to simply allowing all tokens to be considered for merging when the merging ratio surpasses 50%. Here, we present our analysis of adapting PiToMe to our patchified merging technique, evaluated on the Scannet validation set. By fixing a percentage of tokens to remain unmerged (selected using PiToMe's energy score), we merged the remaining tokens, ensuring that the total number of retained tokens (merged and protected) was 20% of the original token count.
>
> | Method                       | mIoU  |
> |------------------------------|-------|
> | Our                           | 77.0  |
> | Our + 1% Protected (PiToMe)   | 76.8  |
> | Our + 5% Protected (PiToMe)   | 76.3  |
> | Our + 10% Protected (PiToMe)  | 75.4  |
>
> Based on the experimental results, our method, by distributing DST sets more evenly across the point cloud to cover a broader range of areas, is more effective than the sparse distribution of DST sets used in PiToMe to maintain a fixed protected set.
>
>
> ## S3DIS Score Clarification
>
> We mistakenly swapped the fine-tuned and off-the-shelf evaluation scores. The corrected results are:
>
> - Fine-tuned: 74.3
> - Off-the-shelf: 72.3
>
> [1]Liu, Zhijian, et al. "Flatformer: Flattened window attention for efficient point cloud transformer." Proceedings of the IEEE/CVF conference on computer vision and pattern recognition. 2023.
> [2]Wang, Peng-Shuai. "Octformer: Octree-based transformers for 3d point clouds." ACM Transactions on Graphics (TOG) 42.4 (2023): 1-11.
> [3]Mao, Yongsen, et al. "SpatialLM: Training Large Language Models for Structured Indoor Modeling." arXiv preprint arXiv:2506.07491 (2025).

---

### Official Review · Reviewer_VXJ3 · 2025-07-03

**Clarity:** 2
**Significance:** 3
**Originality:** 3
**Rating:** 4
**Confidence:** 4

**Summary:**

This paper tackles the problem of token reduction in 3D point cloud transformers by introducing a token merging strategy aimed at significantly lowering computational cost and memory footprint without sacrificing performance. The proposed method builds upon the Point Transformer V3 architecture and incorporates a domain-specific token merging mechanism at each attention block. Central to this approach is the computation of a globally-informed energy score for each point (token) in the input point cloud. This score aggregates the cosine similarity between a token and all patch centroids, effectively capturing its alignment with the global structure. Based on these energy scores, an adaptive token merging strategy is applied inspired by the ToMe framework: tokens within partitions exhibiting lower average energy, indicating weaker alignment with the global structure, are merged more aggressively, while those with higher energy are preserved, controlled by a tunable partition energy threshold. The method is evaluated on two key tasks, indoor semantic segmentation and 3D reconstruction, demonstrating its effectiveness in maintaining performance while significantly improving efficiency.

**Questions:**

- The adaptive merging procedure follows a predefined merging ratio $r$, which determines the number of similar edges to retain in the bipartite graph between source and destination tokens. However, for certain partitions, a different ratio  $r^+>>r$ is used. What is the actual value of this higher ratio, and how is it selected?
- While the discussion on local versus global energy scores in Appendix D is informative, it would be beneficial to include a quantitative comparison to better support the analysis.

**Ethical Concerns:**

["NO or VERY MINOR ethics concerns only"]

**Final Justification:**

The authors have addressed my concerns, particularly regarding the token merging placement and the merging strategy for low- and high-energy tokens. I am raising my rating to borderline accept.

**Limitations:**

Yes

**Paper Formatting Concerns:**

No formatting concerns

**Quality:**

2

**Strengths And Weaknesses:**

**Strengths:**
- The paper provides a systematic analysis of prior token merging methods and offers a clear comparison with the proposed approach.
- A thorough ablation study supports the rationale behind key design choices.
- The proposed method consistently outperforms alternative token merging strategies across all evaluated tasks and matches the accuracy of the original backbone network, while significantly reducing both computational and memory costs.

**Weaknesses:**
- While the overall analysis is solid, some key aspects remain insufficiently explored. First, prior work such as ToMe suggests that token merging is most effective when applied between the attention and MLP layers within each block, and that an alternating strategy for selecting source and destination tokens in the bipartite graph yields better results. In contrast, as shown in Figure 4, the proposed method applies merging before the attention mechanism, and appears to use a random split of tokens within each patch to form the source and destination sets. This deviation from established findings is not clearly justified.
- Second, the logic behind the globally-informed energy score raises concerns. According to lines 194–196 and 199–200, tokens with lower energy, i.e., those that are spatially dissimilar to the overall structure, are aggressively merged, while tokens with higher similarity are preserved. However, within each partition, the merging process appears to follow a different logic: similar tokens are merged, while dissimilar ones are retained. This inconsistency suggests that the adaptive merging strategy may be conceptually reversed. Arguably, partitions with tokens that are aligned with the global structure (and thus potentially redundant) should be merged more aggressively, while partitions with spatially distinct tokens should be preserved to maintain diversity.
- There are structural and editorial inconsistencies throughout the paper. The supplementary material unnecessarily duplicates the entire main paper. Figure 5 presents results for ScanNet200 and S3DIS, yet its caption references additional datasets (nuScenes, Waymo, ScanNet) that are not shown. The second sentence of Table 2’s caption (which is in fact a figure) mistakenly describes Table 1. Figures 1 (main paper) and 10 (supplementary) are duplicates. Additionally, Section 7 of the main paper heavily references analyses from Section B of the supplementary material, which the reader may not have encountered yet, disrupting the flow and self-containment of the discussion.

---

> ### Author Rebuttal · Authors · 2025-07-30
>
> Thank you for your insightful feedback. We will address both of your concerns, including providing additional justification for the choice of token merging and clarifying any confusion regarding the merging algorithm, and incorporate these changes into our manuscript.## Method Justification: Token Merging Placement
>
> Before finalizing our approach, we analyzed the bottleneck in the point cloud transformer model and found that while the attention layer is the primary bottleneck, it is also redundant. In this study, we replicate the analysis of different token merging strategies at a 75% merge rate. For this experiment, we adopt a fixed merging rate rather than a weighted scheme. When merging an MLP layer, we use the output features of the self-attention layer before it as the merging criterion. We evaluated three configurations: merging only the MLP layers (MLP), merging only the attention layers (Attention), and merging both the MLP and attention layers (Attention + MLP). We use Scannet validation set for this experiment.
>
> | Placement             | mIoU  | Latency |
> |-----------------------|-------|---------|
> | PTV3 (Base model)     | 77.7  | 260     |
> | MLP                   | 26.3  | 277     |
> | Attention             | 77.4  | 203     |
> | Attention + MLP       | 25.0  | 212     |
>
> The bottleneck is clearly located in the Attention layer, suggesting that merging at this stage significantly impacts performance. In contrast, the MLP layers, which preserve more detailed information, are critical for maintaining accuracy. Interestingly, increasing computational effort for token merging within the MLP layers led to higher overall computational costs. Based on this analysis, we chose to focus on merging within the Attention layers.
>
>
> ---
>
> ## Conflicting Reasoning
>
> We acknowledge a mistake in the original writing (Line 196). We propose the revised clarification for Lines 196–197:
>
> > "Tokens with lower energy (i.e., **more aligned with global structure**) are considered less informative and can be merged more aggressively, whereas high-energy tokens (i.e., **less aligned with global structure**) are preserved to retain critical information."
>
> ---
>
> ## Writing Improvements
>
> We appreciate the feedback and will revise the manuscript to enhance clarity. Specific improvements include:
>
> - Reorganizing the order and flow of supplementary material.
> - Correcting figure captions and resolving typographical issues (e.g., repeated "the").
>
> ---
>
> ## Merge Rate Selection
>
> We empirically determined a **fixed merging rate of 32** using the validation set of **ScanNet** and applied it across all datasets. Although this fixed rate yields strong performance, we acknowledge the potential benefits of **automatically determining adaptive merge rates per patch**—a promising future direction.
>
> ---
>
>
> ## Qualitative Comparison: Global vs. Local Energy Score
> Thank you for your suggestion. We will include a qualitative comparison between global and local energy scores in our revised version. Below are the comparison results on the ScanNet dataset (without augmentation):
>
> | Merge Rate      | **Global Energy Score** | **Local Energy Score** |
> |--------------------|--------------------------|--------------------------|
> |      Without merging      |       76.3                   |            76.3              |
> |      $ r=0.3$        |       76.4                   |            76.0              |
> |      $ r=0.5$        |        76.2                 |              75.7           |
> |      $ r=0.8$        |      75.8                    |               74.9           |

---

> > ### Comment · Reviewer_VXJ3 · 2025-08-05
> >
> > I appreciate the authors for addressing my concerns, particularly regarding the token merging placement and the clarification of the merging strategy for low- and high-energy tokens. I am raising my rating to borderline accept.

---

### Official Review · Reviewer_7wen · 2025-07-03

**Clarity:** 3
**Significance:** 2
**Originality:** 3
**Rating:** 4
**Confidence:** 3

**Summary:**

This paper adapts and improves token merging techniques from 2D to 3D, demonstrating that 3D point cloud Transformers are heavily over-tokenized. The proposed method is plug-and-play, motivated by the inefficiencies caused by excessive point tokens in both training and inference. However, the merging decisions are based on a fixed heuristic energy score and are only evaluated on indoor benchmarks, raising concerns about the learnability and robustness of the approach in large-scale LiDAR or dynamic scenes.

**Questions:**

see weakness

**Ethical Concerns:**

["NO or VERY MINOR ethics concerns only"]

**Final Justification:**

The authors’ additional experiments have addressed my concerns; I find the paper’s motivation and significance compelling, and therefore, I raise my rating to 4.

**Limitations:**

see weakness

**Quality:**

3

**Strengths And Weaknesses:**

**Strengths**
1. The motivation is valuable, as state-of-the-art 3D scene point cloud methods often require a large number of point tokens to achieve strong performance on both local and global tasks.

2. The modifications proposed by the authors to the PTv3 framework are effective, including the design of a globally structure-aware energy definition, adaptive merging rates, and spatially uniform dst.

**Weaknesses**
1. The use of heuristic energy scores lacks learnability. AdaPT [1], applied in 3D scenes as well, adopts Gumbel-Softmax to enable differentiable token selection and allows dynamic adjustment under computational budgets. However, the current method adopts only two fixed merging ratios, which are neither elegant nor truly dynamic.

2. Several relevant works on token merging or pruning in point clouds have not been discussed, analyzed, or compared, such as YOGO [2], SuperPoint [3], and Token Halting [4]. The current baselines are primarily adapted from 2D methods and directly applied to 3D point clouds, resulting in insufficient baseline comparisons.

3. If further experiments could be conducted in additional domains, such as LiDAR point clouds and object point clouds, it would significantly enhance the impact and credibility of this work.


**Lots of Typos**
1. Line 121      meodels  ->  models
2. Line 165      Sonana and S3DIS  ->  Sonata
3. Line 169      GLOPs  ->  GFLOPs
4. Line 209      differnt dataset  ->  different
5. Line 240      Furthest Point Sampling  ->  Farthest Point Sampling
6. Figure 5       nuScenese  ->  nuScenes
7. Figure 5       Wamo  ->  Waymo




[1] Adaptive Point Transformer

[2] You Only Group Once: Efficient Point-Cloud Processing with Token Representation and Relation Inference Module

[3] Efficient 3D Semantic Segmentation with Superpoint Transformer

[4] Efficient Transformer-based 3D Object Detection with Dynamic Token Halting

---

> ### Author Rebuttal · Authors · 2025-07-30
>
> ## Lacking Learnability
>
> We agree that while our proposed globally-informed graph provides a way to determine which areas contain more redundant information (and can thus be processed with more aggressive merging), automatically determining the merging rate for each region is indeed a promising next step in our research direction. We view this as a natural extension to make our method more adaptive and fully learnable.
>
> ---
>
> ## Missing Comparison
>
> We appreciate the suggestion to compare with other recent token pruning and downsampling strategies. Below, we discuss relevant methods:
>
> - **YOGO**
>   YOGO performs cross-attention between regions and original points to enable a wider receptive field. However, it relies on furthest point sampling, which is computationally expensive for large scenes with many points and requires many point groups. Additionally, it requires training from scratch. As an attempt to compare, we adopt their grouping strategy in our ablations.
>
> - **TokenHalting**
>   This method determines which token to prune at each layer using a learnable criterion. At the final detection head, pruned features are restored for full processing. However, this approach is primarily suitable for architectures without explicit spatial downsampling. It is less applicable to our architecture, which benefits from spatial relationships.
>
> - **SuperPoint**
>   This method proposes a new architecture that gradually groups points into super points. Self-attention is then performed on aggregated features within each group. While efficient, it lacks fine-grained point-level processing and depends on hand-crafted features. It also requires training from scratch.
>
> Our method instead targets redundancy within general-purpose architectures such as UNet and Point Transformer (PTv3, Sonata). These architectures already demonstrate strong performance across both indoor (ScanNet) and outdoor (LiDAR) point cloud benchmarks. Our token merging strategy can be applied off-the-shelf, improving efficiency **without retraining**, which contrasts with the above methods that require full retraining or architectural changes.
>
> We also tried to adopt YoGo by directly integrating their downsampling methods into the PTv3 model and performing off-the-shelf inference on the ScanNet dataset. we simulated the cross-attention mechanism in two steps: (1) pooling $n$ Key and Value ($K, V$) tokens into $k$ centroids using a kernel size of 64 and a stride of 32 to mimic self-attention across regions, and (2) performing self-attention between the $n$ Query ($Q$) tokens and the $k$ pooled $K, V$ tokens, thus simulating cross-attention between the original points and the regions.
>
> | Method        | mIoU | mAcc | allAcc |
> |---------------|------|------|--------|
> | PTv3          | 77.6 | 87.2 | 94.6   |
> | PTv3 + YoGo   | 66.5 | 72.2 | 86.6   |
> | PTv3 + Ours   | 77.0 | 86.6 | 92.6
> ---
>
> ## Experiment with LiDAR Point Cloud and Object Point Cloud
>
> For experiments with LiDAR point clouds, we include results on **nuScenes** [2] in **Tab. 7** of the supplementary material. We report the relevant summary here:
>
> | Method        | mIoU | mAcc | allAcc | peakMem (GB) | GFLOPS | Latency (ms) |
> |---------------|------|------|--------|---------|-------|---------|
> | PTv3          | 80.3 | 87.2 | 94.6   | 6.20|  101.68     |   152      |
> | PTv3 + Ours   | 78.0 | 85.5 | 94.0   | 0.92 |32.45|   106      |
>
> ---
>
> We also evaluate our method with a recent object point cloud segmentation model, **SAMPart3D** [1], on their provided dataset:
>
> | Method              | Human-Shape | Animals | Daily-Used | Buildings&&Outdoor | Transportations | Plants  | Food    | Electronics | Total Mean AP | Peak GPU Mem |
> |---------------------|-------------|---------|------------|--------------------|------------------|---------|---------|-------------|----------------|----------------|
> | **SamPart3D**        | 0.4454      | 0.4224  | 0.3911     | 0.3109             | 0.3683           | 0.4466  | 0.4693  | 0.3576      | 0.3904         | 3.44GB         |
> | **SamPart3D + Ours** | 0.4182      | 0.4194  | 0.3782     | 0.2895             | 0.3329           | 0.4113  | 0.4670  | 0.3955      | 0.3785         |1.45GB         |
>
> We applied a default merge rate of 0.6 and an aggressive merge rate of 0.9 for the experiments. We did not have enough time to complete our analysis for the GFLOPs in this experiments. However, we reported a significant drop of memory consumption from 3.44GB to 1.45GB while only have 2% drop in total AP score. This suggest that we would likely observe a huge drop rate with GFLOPs. Fine-tunning experiment on this dataset is also promising.
> ---
>
> Additionally, we assess our method on the **SpatialLM** dataset for object detection using the **SpatialLM** 3D language model:
>
> | Configuration   | F1_Layouts_25 | F1_Layouts_50 | F1_Objects_25 | F1_Objects_50 | Avg_Inference_Time | Peak_Memory_GB |
> |-----------------|---------------|---------------|---------------|---------------|--------------------|-----------------|
> | baseline        | 0.4906        | 0.3886        | 0.3356        | 0.1894        | 6.009              | 12.36           |
> | weighted_0.8    | 0.4809        | 0.3873        | 0.3485        | 0.2006        | 4.795              | 2.53            |
>
> The metrics represent the F1 score of object detection at IoU thresholds of 25 and 50. In the language and 3D-grounded domains, our results show that the tokens in the point cloud transformer remain highly redundant. Our method therefore can significantly boost the computational performance in term of inference speed and peak GPU memory consumption.
>
> ---
>
> We hope these comparisons and extensions clarify the generality and applicability of our method across different domains and architectures.
>
> [1] Mao, Yongsen, et al. "SpatialLM: Training Large Language Models for Structured Indoor Modeling." arXiv preprint arXiv:2506.07491 (2025).

---

> ### Comment · Reviewer_7wen · 2025-08-06
>
> I appreciate the authors' rebuttal, which was convincing. I will raise my score.

---

> > ### Author Response · Authors · 2025-08-09
> > **Thank you Reviewer 7wen**
> >
> > Thank you very much for taking the time to review our rebuttal and for raising your score. As suggested, we will incorporate all additional results into the revised version.

---

### Official Review · Reviewer_vPay · 2025-07-03

**Clarity:** 3
**Significance:** 3
**Originality:** 3
**Rating:** 4
**Confidence:** 5

**Summary:**

This research highlights the importance of bringing the idea of "patchify" to point cloud transformers, similar to how it's used in image transformers. The motivation is clear: there are just too many tokens to process, which holds back progress in point cloud processing. But applying "patchify" to sparse point clouds (without losing too much information) isn't as straightforward as it is with images. This paper tackles that challenge and proposes a method that strikes a solid balance between accuracy and token count.

**Questions:**

See the discussion in the Weakness section. I initially suggested a `borderline accept` and am open to adjusting my score based on the authors’ rebuttal.

**Ethical Concerns:**

["NO or VERY MINOR ethics concerns only"]

**Final Justification:**

The author's rebuttal solved my concern. Therefore, I maintain my recommendation score as "weakly accept". Overall, I support the acceptance mostly because I think this is a good direction, worth encouraging the point cloud community to pay more attention to this direction. The manuscript, maybe without too much technical contribution, yet does provide a simple yet effective baseline in this direction.

**Limitations:**

Yes

**Paper Formatting Concerns:**

No major formatting issues are found.

**Quality:**

2

**Strengths And Weaknesses:**

## Strengths
**[Good Motivation / Direction]**
Overall, I like this paper’s starting point and think it's doing some of the right things for our community. We've already put a lot of effort into improving the efficiency and scalability of basic point cloud operators. But no matter how far we push those boundaries, the most straightforward way to reduce cost is to process fewer points. That’s why we need the concept of "patchifying", similar to what’s been done in image processing. In that sense, the proposed "merging" techniques make a lot of sense.

**[Performance]**
The accuracy from the merged point cloud reported by the paper is quite acceptable. Compared with the compression ratio, I think it would be helpful to some applications in the robotic learning or autonomous driving area. And also, it is good to extend the downstream tasks from perception to reconstruction.

## Weakness
**[Efficiency Analysis of Patchifying Itself]**
I understand there's no doubt that processing a patchified point cloud is more efficient. However, I'm also curious about the efficiency of the patchifying algorithm itself. I think the paper would be stronger if it included some latency benchmarks comparing the different patchifying methods discussed.

**[Absence of ScanNet Semantic Segmentation]**
I noticed the paper includes semantic results on ScanNet200, so it feels a bit odd that ScanNet itself isn’t included (especially since the main difference is just the label mapping). It would be helpful to at least discuss performance on ScanNet, even if the results aren’t great.

**[Please Polish the TeX (Minor)]**
I’d suggest the authors double-check the consistency of terminology used throughout the paper. For example, I noticed that Point Transformer V3 is referred to in multiple ways, such as PTv3, PvT-3, and Point Transformer v3. It would be better to stick to a single, consistent naming convention.

---

> ### Author Rebuttal · Authors · 2025-07-30
>
> Thank you for your valuable feedback and suggestions. We provide detailed clarifications below.
>
> ## ScanNet Results
>
> The evaluations for the validation sets of both **ScanNet** and **ScanNet200** are included in **Tab. 1** of the manuscript. The results are illustrated in the first and second columns, respectively.
>
> ---
>
> ## Latency Comparison
>
> We analyze latency on the validation scenes of **ScanNet** and compare various grouping strategies. For each method, we adjust its parameters so that the number of points retained is around 20% of the original point cloud. The results are summarized below:
>
> | **Method**            | **mIoU (%)** | **Latency (ms)** |
> |-----------------------|--------------|-------------------|
> | PTv3 (base model)           | 77.6        | 266                 |
> | Random Drop           | 70.1         |          154        |
> | FPS                   | 71.2         | 3122                 |
> | Voxel Downsampling    | 72.1         | **160**                 |
> | Progressive           | 75.7         | 198                 |
> | PoolTome              | 71.0         | 194                 |
> | **Ours**              | **77.0**     | 203                 |
>
> ---
>
> We introduce several additional grouping techniques for token reduction (for more information please see **Sec. B** and **Tab. 5** in the supplementary material):
>
> - **Progressive-Tome**
>   We design an $O(N)$ token merging method by constructing a local graph among tokens, where edges are formed between adjacent tokens in a 1D serialized order. In our experiments, we merge the top 80% of edges with the highest similarity by averaging their corresponding tokens. The unmerging process follows the same approach as our main method.
>
> - **PoolTome**
>   To enhance spatial communication, we first apply average pooling with a large kernel size and stride, both set to $\log_2(N)$, where $N$ is the number of tokens. Attention is then computed over the pooled features. Finally, unpooling is performed by duplicating the attention outputs back to their corresponding original token positions. The overall attention computation cost is $O(\log_2(N))$. This method can be considered a token merging strategy that relies purely on locality.
>
> ---
>
> While point cloud downsampling techniques such as Random Drop and Voxel Downsampling offer computational efficiency, they are often challenging to apply in off-the-shelf scenarios, as they typically overlook both point-wise feature representations and the latent space of segmentation models. In contrast, our method identifies redundancy using a globally informed graph structure, enabling effective merging and unmerging of points while preserving the features essential for accurate model performance.

---

> ### Comment · Reviewer_vPay · 2025-08-06
>
> The author's rebuttal addressed my concerns regarding accuracy and efficiency. The additional discussion on PTv3 and Sonata is also appreciated. Therefore, I maintain my recommendation as `weakly accept`. Overall, I support acceptance primarily because I believe the proposed direction has its value and merits further attention from the point cloud community. While the manuscript may not present substantial technical novelty, it introduces a simple and effective baseline that could stimulate progress in this area.
>
> PS: Regarding the discussion of Sonata (self-supervised pre-training) from the author’s response to other reviewers: I would like to share my perspective. The purpose of large-scale pre-training is not necessarily to improve the efficiency of pre-training itself, but rather to enhance the efficiency of downstream fine-tuning—regardless of data or computational resources used during pre-training (as seen in methods like DINO or CLIP). Thus, the heavy use of resources in pre-training should not be viewed as an inherent drawback. That said, efficiency in pre-training and efficiency in fine-tuning are not mutually exclusive. *In fact, it would strengthen the manuscript if the authors could demonstrate that fine-tuning from well-pretrained weights (e.g., Sonata) combined with their proposed method leads to improved efficiency*—perhaps using the ScanNet data efficiency benchmark. I hope the authors consider this as an optional suggestion for revision.

---

> ### Author Response · Authors · 2025-08-09
> **Efficient Downstream Task Finetuning for Sonata**
>
> Thank you for your interesting suggestion. Combining our technique with well-pretrained weights (e.g., Sonata) for downstream task training is a promising and impactful direction. Despite the limited time available for the rebuttal, we have attempted to conduct a preliminary experiment to explore this.
>
> In this experiment, we fine-tuned Sonata on the **ScanNet** dataset using our method with a **70% merging rate**. This setup differs from the experiments in the main paper in the following ways:
>
> - **Main paper**:  Fine-tuned from a checkpoint where the downstream task had already been trained from pretrained Sonata *(Self-pretrained Sonata → Downstream task training → Fine-tuning with our method)*.
>
>
> - **Rebuttal experiment**: Trained the downstream task directly from pretrained Sonata *(Self-pretrained Sonata → Downstream task training  combined with our method)*.
>
>
> ---
>
> ## Results Summary
>
> ### 1. Linear Probing
> In this setting, the **encoder was frozen**, and we trained a **smaller decoder** (a linear probing).
> - **Setup**: batch size = 12, 1 GPU: H100
> - **Row 1**: Results from the original paper.
> - **Row 2**: Our attempt at training the downstream task with token merging (**merge 70% tokens**).
>
> | Version              | mIoU  | mAcc  | allAcc | Peak GPU Mem | GPU hours |
> |----------------------|-------|-------|--------|--------------|---- |
> | Sonata-lin           | 72.52 | 83.11 | 89.74  | 50.2 GB  |   16.2   |
> | Sonata-lin + Ours    | 72.43 | 83.31 | 89.71  | 45.8 GB |    11.2  |
>
> ---
>
> ### 2. Decoder Fine-tuning
> In this setting, both Encoder and Decoder are updated, we trained this setting using 75% number of epochs due to time constraints.
> - **Setup**: Fine-tune Sonata with a lightweight decoder (batch size = 16, H100 GPUs).
> - **Row 1**: Results from the original paper.
> - **Row 2**: Our results using token merging (**merge 70% tokens**).
>
> | Version                  | mIoU  | mAcc  | allAcc | Peak GPU Mem |GPU hours|
> |--------------------------|-------|-------|--------|--------------|--- |
> | Sonata-ft          | 79.0  | 86.6  | 92.7   | 211.25 GB     |  55.2  |
> | Sonata-ft + Ours    | 78.6  | 86.4  | 92.3   | 74.95 GB|  28.3 |
>
> ---
>
>
>
> ## Conclusion
>
> In both setting the performance remains nearly identical, while GPU memory usage and training time drops significantly. This demonstrates that our technique can enable efficient downstream training, not just fine-tuning pre-trained downstream task models.
>
> While this experiment is limited in scope due to time constraion (one dataset and one merge rate ), it provides **initial evidence** that Sonata with 70% token merging **preserves features sufficiently for downstream training**.  This also suggests **promising results** for efficient downstream task training of large models (e.g., Sonata with a large decoder and full fine-tuning).  We plan to extend this experiment to larger-scale settings and will report results in the revision.

---

### Note · Authors · 2025-08-13

Reviewers agree that our paper addresses a critical problem: scaling transformer-based 3D models by reducing token processing, and that our globally-informed, geometry-aware token merging strategy is both practical and effective. The strong empirical validation across multiple tasks and datasets highlights its ability to achieve substantial computational savings with minimal or no accuracy loss, occasionally outperforming the baseline.

We appreciate the reviewers’ feedback, especially the recognition of our token merging strategy. Reviewers 7wen and VXJ3 raised their scores after our clarifications on ambiguities, generalization to diverse data types, additional comparison with baselines and additional ablation studies. We hope reviewers cv9K and vPay will carefully consider our detailed responses.

### **Responses to remaining concerns**

**Generalization to Different Data Types (Reviewers cv9K, 7wen)**

We strengthen our generalization claims by presenting experiments on diverse datasets (nuScenes, SAMPart3D, SpatialLM), demonstrating robustness across various settings, from indoor scenes to autonomous driving.

**Efficient Downstream Training (Reviewer vPay)**

We show that our method integrates seamlessly into downstream task training pipelines, significantly reducing computational costs while maintaining performance - showing is a promising exploration.

**Theoretical Foundation (Reviewer cv9K)**

- Our work is the first to systematically identify and quantify token redundancy in large-scale 3D transformers (e.g., Ptv3, Sonata). While formal proofs are valuable, our strong empirical results clearly demonstrate the significance of our approach.
- Transfer Entropy Analysis: We conducted a transfer entropy analysis on ScanNet, showing minimal information loss after aggressive token merging, further supporting our claim of significant redundancy in point transformer tokens. This theoretical framework supports our claim of token redundancy and validates our Global Informed Graph's effectiveness.

**Novelty (Reviewer cv9K)**
We emphasize that our method is fundamentally different from RandLA-Net. While RandLA-Net is an independent model that uses random sampling for downsampling, our approach is a plug-and-play strategy that reduces redundancy through feature similarity and adaptive, globally-informed scoring. Furthermore, we present side-by-side comparisons with an adapted RandLA-Net, demonstrating that our method achieves superior performance.

---

### Decision · Program_Chairs · 2025-09-17

**Decision:**

Accept (poster)

**Comment:**

This paper proposes token merging techniques for 3D point cloud transformers, showing that existing 3D transformer architectures are heavily over-tokenized. The main strength of the paper is the effectiveness of the proposed modifications to the PTv3 framework -- the method can even surpass the baseline by fine-tuning the strategy while removing 90–95% of the tokens.

Several weaknesses were identified, such as the absence of latency benchmarks, limited discussion of other token-merging/pruning techniques (e.g., Yogo), limited experiments on LiDAR point clouds, missing adaptations of other token-merging strategies, reliance on heuristic energy scores, insufficient theoretical justification, and questions about novelty relative to existing methods (e.g., RandLA-Net).

The authors provided extensive clarifications, including latency experiments, comparisons with other merging techniques, additional results on LiDAR point clouds, adaptations of alternative token-merging strategies. Four out of five reviewers were satisfied with the responses and voted for acceptance. One reviewer, however, remained negative, expressing concerns about novelty relative to RandLA-Net and claiming limited empirical generalization and theoretical justification. The AC finds that these concerns were not sufficiently substantiated by the reviewer. The authors presented experiments on diverse datasets (nuScenes, SAMPart3D, SpatialLM), demonstrating robustness across various settings, from indoor scenes to autonomous driving. The AC agrees with the majority of reviewers that the method is sufficiently distinct from prior work (including RandLA-Net) and provides adequate intuition for its design choices.

Thus, the AC recommends acceptance. The authors are strongly encouraged to incorporate the additional experiments, results, and clarifications from the rebuttal and discussion into the final version of the paper.